



# A comprehensive in-situ and remote sensing data set collected during the HALO–$(\mathcal{AC})^3$ aircraft campaign

André Ehrlich[1], Susanne Crewell[2], Andreas Herber[3], Marcus Klingebiel[1], Christof Lüpkes[1], Mario Mech[2], Sebastian Becker[1], Stephan Borrmann[4,5], Heiko Bozem[4], Matthias Buschmann[6], Hans-Christian Clemen[5], Elena De La Torre Castro[7, 4, 8], Henning Dorff[9], Regis Dupuy[10], Oliver Eppers[5], Florian Ewald[7], Geet George[11,12], Andreas Giez[13], Sarah Grawe[14], Christophe Gourbeyre[10], Jörg Hartmann[3], Evelyn Jäkel[1], Philipp Joppe[4,5], Olivier Jourdan[10], Zsófia Jurányi[3], Benjamin Kirbus[1], Johannes Lucke[7, 8], Anna E. Luebke[1], Maximilian Maahn[1], Nina Maherndl[1], Christian Mallaun[13], Johanna Mayer[7], Stephan Mertes[14], Guillaume Mioche[10], Manuel Moser[7, 4], Hanno Müller[1], Veronika Pörtge[15], Nils Risse[2], Greg Roberts[16], Sophie Rosenburg[1], Johannes Röttenbacher[1], Michael Schäfer[1], Jonas Schaefer[14], Andreas Schäfler[7], Imke Schirmacher[2], Johannes Schneider[5], Sabrina Schnitt[2], Frank Stratmann[14], Christian Tatzelt[14], Christiane Voigt[7, 4], Andreas Walbröl[2], Anna Weber[15], Bruno Wetzel[14], Martin Wirth[7], and Manfred Wendisch[1]

[1]Leipziger Institut für Meteorologie (LIM), Universität Leipzig, Leipzig, Germany
[2]Institut für Geophysik und Meteorologie (IGM), Universität zu Köln, Cologne, Germany
[3]Alfred–Wegener–Institut, Helmholtz–Zentrum für Polar– und Meeresforschung (AWI), Bremerhaven & Potsdam, Germany
[4]Institut für Physik der Atmosphäre (IPA), Johannes Gutenberg-Universität, Mainz, Germany
[5]Particle Chemistry Department, Max Planck Institute for Chemistry (MPIC), Mainz, Germany
[6]Institute of Environmental Physics, University of Bremen, Bremen, Germany
[7]Institute for Physics of the Atmosphere, Deutsches Zentrum für Luft- und Raumfahrt, Wessling, Germany
[8]Faculty of Aerospace Engineering, Delft University of Technology, Delft, Netherlands
[9]Meteorologisches Institut, Universität Hamburg, Hamburg, Germany
[10]Laboratoire de Météorologie Physique (LaMP), Université Clermont Auvergne/ OPGC/CNRS, UMR 6016, Clermont-Ferrand, France
[11]Max Planck Institute for Meteorology, Hamburg, Germany
[12]Faculty of Civil Engineering and Geosciences, Delft University of Technology, Delft, Netherlands
[13]German Aerospace Center, Flight Experiments, Oberpfaffenhofen, Germany
[14]Leibniz–Institut für Troposphärenforschung (TROPOS), Leipzig, Germany
[15]Meteorologisches Institut, Ludwig-Maximilians-Universität München, Munich, Germany
[16]Scripps Institution of Oceanography, UC San Diego, USA

**Correspondence:** André Ehrlich (a.ehrlich@uni-leipzig.de)

**Abstract.**

The HALO–$(\mathcal{AC})^3$ aircraft campaign was carried out in March and April 2022 over the Norwegian and Greenland Seas, the Fram Strait, and the central Arctic Ocean. Three research aircraft, the High Altitude and Long Range Research Aircraft (HALO), Polar 5, and Polar 6, performed 54 partly coordinated research flights on 23 flight days over areas of open ocean, the marginal sea ice zone (MIZ), and the central Arctic sea ice. The general objective of the research flights was to quantify the evolution of air mass properties during moist and warm air intrusions (WAIs) and cold air outbreaks (CAOs). To gain a comprehensive data set, the three aircraft followed different strategies. HALO was equipped with active and passive remote





sensing instruments and dropsondes to cover the regional evolution of cloud and thermodynamic processes. Polar 5 carried a similar remote sensing payload as HALO, and Polar 6 was instrumented with in-situ cloud, aerosol, and trace gas instruments
focusing on the initial air mass transformation close to the MIZ. The processed, calibrated, and validated data are published in the World Data Center PANGAEA as instrument-separated data subsets and listed in aircraft-separated collections for HALO (Ehrlich et al., 2024a), Polar 5 (Mech et al., 2024a), and Polar 6 (Herber et al., 2024). A detailed overview of the available data sets is provided here. Furthermore, the campaign-specific instrument setup, the data processing, and data quality are summarized. Based on measurements conducted during a specific CAO, it is shown that the scientific analysis of the HALO–
$(\mathcal{AC})^3$ data benefits from the coordinated operation of the three aircraft.

# 1   Introduction

The Arctic currently experiences a strong warming that exceeds global warming by a factor of two to four (Rantanen et al., 2022). In addition to the remarkable sea ice retreat, the enhanced Arctic warming represents one of the most obvious results of Arctic-specific processes and feedback mechanisms causing significant climate changes in the Arctic that are widely referred to
as Arctic amplification (Serreze et al., 2009). The effects and consequences of Arctic amplification are extensively discussed in literature (e.g., Wendisch et al., 2023a). However, especially the role of clouds and air mass transports into and out of the Arctic and related transformation processes are still poorly understood. Therefore, in spring 2022 a major observational initiative, the HALO–$(\mathcal{AC})^3$ campaign, was conducted (Wendisch et al., 2021, 2024). HALO–$(\mathcal{AC})^3$ combined airborne, balloon-borne, and ground-based measurements aiming to provide detailed observations of thermodynamic, cloud, aerosol, trace gas, and
surface properties as they change during the meridional air mass transports. Furthermore, the collected data will be compared to simulations conducted with numerical Arctic weather and climate models.

The focus area of interest of HALO–$(\mathcal{AC})^3$ was the North Atlantic corridor of the Fram Strait between Greenland and Svalbard, which is a major entrance/exit gate for air masses moving North or South between mid and high latitudes also referred to as North Atlantic pathway (Mewes and Jacobi, 2019; Dahlke et al., 2022; Papritz et al., 2022). In this region,
warm and moist mid-latitude air masses are pushed frequently northwards into the sea ice-covered central Arctic (Pithan et al., 2018; Liang et al., 2023). Cold air masses moving southwards regularly form cold air outbreaks (CAOs) over the open ocean (Dahlke et al., 2022). Due to the seasonally relatively stable sea ice edge in the Fram Strait and the permanent measurements on Svalbard, in particular at Ny-Ålesund research stations, this region has become a hot spot of Arctic research.

A series of airborne, ship, and ground-based campaigns, such as ASTAR, ACLOUD, AFLUX, MOSAiC-ACA, PAMAR-
CMiP, ASCOS, Arctic Ocean, COMBLE, NASCENT (Wendisch et al., 2019; Mech et al., 2022a; Tjernström et al., 2014; Vüllers et al., 2021; Geerts et al., 2022; Pasquier et al., 2022), have provided valuable data to enhance our understanding of the Arctic climate system and improve numerical models (e.g., McCusker et al., 2023). However, most of the previous airborne campaigns in the Fram Strait were limited to smaller areas due to restricted flight ranges of the involved planes, or were focused on atmospheric processes over open ocean, e.g., the downstream development of CAOs (Vüllers et al., 2021). Observations
from ships such as MOSAiC (Shupe et al., 2022) provide longer term measurements in the Arctic sea ice but are mostly re-





stricted to the local scale. The permanent research stations at the ground are often affected by specific orographic effects and not always representative for the Arctic Ocean (Gierens et al., 2020). Therefore, the HALO–$(\mathcal{AC})^3$ aircraft campaign was designed to combine different aircraft platforms and aiming for observations in a quasi-Lagrangian approach.

To characterize the process of air mass transformation and to quantify their consequences, measurements of thermodynamic parameters, cloud and aerosol particles properties, trace gas concentrations, and the radiative energy budget are most relevant. Especially close to the sea ice edge, these transformations can be fast (Murray-Watson et al., 2023; Kirbus et al., 2024) requiring small-scale observations. Therefore, three research aircraft equipped with state-of-the-art instrumentation were involved in HALO–$(\mathcal{AC})^3$ to sample atmospheric parameters on different spatial scales and during different stages of air mass transformation (Wendisch et al., 2024).

The observations cover major warm air intrusions (WAIs) injecting warm and moist air into the central Arctic. Multiple cold air outbreaks (CAOs) were characterized in their initial stage close to the sea ice edge with the Polar 5 and Polar 6 research aircraft and in a quasi-Lagrangian perspective jointly with HALO. This approach allows to quantify the air mass transformations by deriving changes of thermodynamic profiles, large scale subsidence, and cloud properties over a period of 24 hours. Individual events of high latitude Arctic cirrus and the formation of a polar low are included in the data set.

The collected airborne data set includes a multitude of in-situ and remote sensing instruments and provides measurements of basic thermodynamic quantities, the radiative energy budget and cloud, aerosol, and trace gas properties. The aim of this paper is to give an overview of the available data sets of the airborne measurements, their accessibility, and potential uncertainties that have to be considered when using the data. In Section 2 the general setup and an overview of the HALO–$(\mathcal{AC})^3$ aircraft activities are summarized. The data sets and their availability are described in Sections 3 and 4. To highlight the potential of merging the data of the different aircraft, Section 5 outlines examples of combined data analysis on the basis of one case study.

## 2 HALO–$(\mathcal{AC})^3$ aircraft campaign

### 2.1 General setup

HALO–$(\mathcal{AC})^3$ was designed as a concerted effort combining airborne, ground-based, and balloon-borne observations in area around Svalbard and in the central Arctic. For the aircraft campaign in March and April 2022, the German High Altitude and Long Range Research Aircraft (HALO) operated by the German Aerospace Center (DLR, Krautstrunk and Giez, 2012) was equipped with a comprehensive suite of active and passive remote sensing instruments and dropsondes (Stevens et al., 2019). Based in Kiruna, Sweden, HALO with its extended flight range covered one of the major sectors of WAIs and CAOs (Mewes and Jacobi, 2019; Liang et al., 2023) and followed air masses into the central Arctic close to the north pole (8000 km range in altitudes up to 13 km). The Polar 5 (remote sensing) and Polar 6 (in-situ) aircraft of the Alfred Wegener Institute (AWI, Wesche et al., 2016) probed the lower troposphere out of Longyearbyen, Svalbard (1500 km range in altitudes up to 4 km). Both aircraft mostly operated in the Fram Strait with a focus on the local transitions at the sea ice edge. Several coordinated flights between the three aircraft were conducted with Polar 6 sampling in-situ aerosol, cloud, and precipitation particles within





the boundary layer, Polar 5 observing clouds and precipitation from above roughly at 3 km altitude, and HALO providing the large scale view and following air masses during one single and along successive flights.

During the HALO–$(\mathcal{AC})^3$ campaign period, an intensive atmospheric measurements program extending the permanent observations was conducted at the research base AWIPEV at Ny-Ålesund/Svalbard (78° 55' N, 11° 55' E, Neuber, 2006) jointly operated by AWI and the French Polar Institute Paul-Émile Victor (IPEV). The tethered balloon BELUGA (Balloon-born moduLar Utility for profilinG the lower Atmosphere) profiled the atmospheric boundary layer and the lower free troposphere from the ground to about 1 km height with turbulence, radiation, and aerosol particle probes (Egerer et al., 2019; Lonardi et al.,

2024). During March and April 2022, the frequency of radiosonde launches at AWIPEV was increased to six-hourly intervals (Maturilli, 2022a, b). To link the airborne measurements with the observations at AWIPEV, Polar 5 and Polar 6 regularly passed over the site during the outbound or inbound flights.

    In parallel to HALO–$(\mathcal{AC})^3$, the ISotopic Links to Atmospheric water's Sources (ISLAS) campaign and the airborne field campaign Arctic Cold Air Outbreak (ACAO) were performed. For ISLAS, the ATR-42 research aircraft operated by SAFIRE

(french: Service des Avions Français Instrumentés pour la Recherche en Environnement, Lamorthe et al., 2016) was deployed with in-situ cloud, aerosol, and trace gas probes and remote sensing instruments. ACAO used the BAE-146 operated by the Facility for Airborne Atmospheric Measurements (FAAM) to measure cloud, aerosol and atmospheric quantities. Both aircraft were stationed in Kiruna and mostly operated in low altitudes between Svalbard and Northern Scandinavia. To combine the efforts, HALO, ATR-42, and BAE-146 were coordinated in four flights.

## 2.2   Overview of flight activities

An overview of all HALO–$(\mathcal{AC})^3$ research flight activities including information on the coordination and dropsondes is provided in Table 1, the flight tracks are illustrated in Fig. 1. In total, HALO performed 17 research flights (RF), while Polar 5 and Polar 6 conducted 13 flights each. Nine HALO flights were coordinated with Polar 5 and/or Polar 6. For close spatial and temporal coordination of the three aircraft, a standard leg (SL) between two fixes way points (79° 24' N, 7° 37' E and

79° 56' N, 1° 31' E) was defined and repeatedly implemented in the flight planning. This straight SL connects areas of open ocean to the East and the dense sea ice further west. Most of the time, the SL crossed the main direction of air mass advection in the boundary layer (Walbröl et al., 2023) and, therefore, allowed for detailed sampling of air masses crossing the sea ice edge. Four HALO flights were coordinated with the ATR-42 and/or BAE-146 aircraft providing additional in-situ cloud observations. For all research flights, individual flight reports were compiled by the team of participants and are published by Ehrlich

et al. (2024). These reports summarize the general objective of the research flights, motivate the flight pattern in combination with the current synoptic weather conditions, and provide notes by the instrument operators, quick-looks, and images from the flights.

    In total, the measurement time of HALO amounts to 143 hours covering a distance of 113,000 km. Polar 5 and Polar 6 operated for about 53 hours and 64 hours, respectively. Due to the lower flight speed of the Polar 5 and 6 aircraft, this corre-

sponds to a distance of 36,000 km for each aircraft. The spatial distribution of measurements is illustrated in Fig. 1 and shows a focus of measurements around the sea ice edge in the Fram Straight. This mostly results from Polar 5 and Polar 6, but also

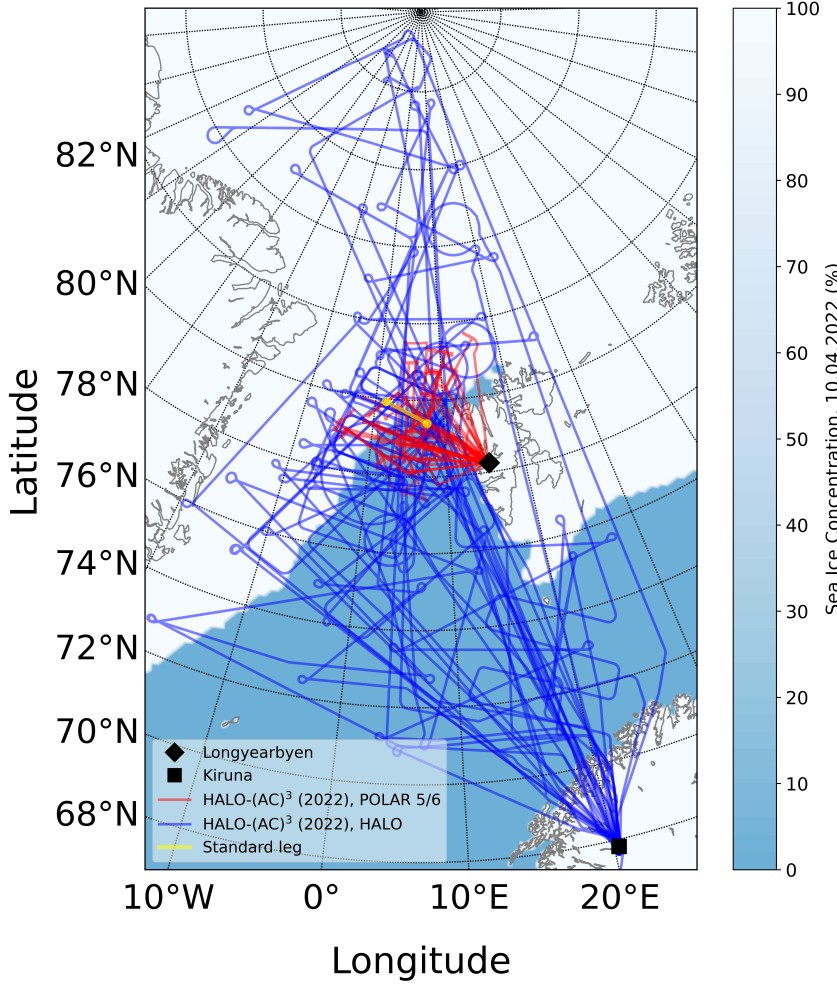

**Figure 1.** Distribution of the flight tracks of all aircraft, Polar 5 and Polar 6 (red), and HALO (blue) during HALO–$(\mathcal{AC})^3$. The sea ice cover corresponds to the middle of the campaign period 30 March 2022. The yellow line indicates the standard flight leg.

from the intense coordination with HALO in this region. The SL features the highest density of measurement, it was flown in collocation during five flights. The flights covered different surface conditions, including snow-covered land, open ocean, closed sea ice, and the MIZ as summarized for each aircraft in Fig. 2. The classification uses satellite observations of sea ice

concentrations provided by the Advanced Microwave Scanning Radiometer (AMSR-II, Spreen et al., 2008). The definition of the MIZ follows the common thresholds of sea ice concentration between 15 % and 80 % (Strong et al., 2017). With above 50 % of the flight time, open ocean is the dominant surface type observed by the three aircraft. Due to the longer endurance, HALO covered a higher fraction of observations over sea ice (30 % of total flight time) than Polar 5 and Polar 6 (20 %). The MIZ was sampled more intensively with Polar 5 and Polar 6. A similar amount of flight time of Polar 5 and Polar 6 was spent

over land to coordinate with the columnar observations performed at AWIPEV in Ny-Ålesund. Vertically, the flight altitudes of



**Table 1.** Overview of HALO–$(\mathcal{AC})^3$ research flights including the research flight (RF) number and information on the coordination of HALO with the Polar 5 (P5), Polar 6 (P6), BAE-146, and ATR-42 aircraft. Flights covering the standard leg (SL) and overpasses of Polar 5 over AWIPEV station in Ny-Ålesund (NyÅ) are indicated. The number of successfully launched dropsondes from Polar 5 and HALO is given. For HALO dropsondes, the number of launches used in the Global Telecommunication System (GTS) data assimilation is specified in brackets.

| Day | Research flight (RF) | | | Coordination | Standard | AWIPEV | Dropsondes from | |
|-----|------|---------|---------|-----------------|------|--------|-----|------------|
|     | HALO | Polar 5 | Polar 6 | HALO perspective | Leg |        | P5  | HALO (GTS) |
| 12.03.2022 | 02 | - | - |  |  |  | - | 20 |
| 13.03.2022 | 03 | - | - |  |  |  | - | 21 |
| 14.03.2022 | 04 | - | - |  |  |  | - | 9 |
| 15.03.2022 | 05 | - | - |  |  |  | - | 25 (3) |
| 16.03.2022 | 06 | - | - | BAE-146 |  |  | - | 23 (19) |
| 20.03.2022 | 07 | 01 | 01 | P5, P6 |  | NyÅ | 12 | 17 (13) |
| 21.03.2022 | 08 | - | - | BAE-146 |  |  | - | 13 (13) |
| 22.03.2022 | - | 02,03 | 02 |  |  | NyÅ | 12 | - |
| 24.03.2022 | - | - | 03 |  |  |  | - | - |
| 25.03.2022 | - | 04 | - |  |  | NyÅ | 5 | - |
| 26.03.2022 | - | - | 04 |  |  |  | - | - |
| 28.03.2022 | 09 | 05 | 05 | P5, P6 | SL | NyÅ | 15 | 16 (16) |
| 29.03.2022 | 10 | 06,07 | 06 | P5, P6, ATR-42, BAE-146 |  | NyÅ | 5 | 18 (10) |
| 30.03.2022 | 11 | 08 | 07 | P5, P6, ATR-42, BAE-146 |  | NyÅ | 15 | 32 (32) |
| 01.04.2022 | 12 | 09 | 08 | P5, P6 | SL | NyÅ | 18 | 41 (41) |
| 04.04.2022 | 13 | 10 | 09 | P5, P6 | SL |  | 14 | 13 (11) |
| 05.04.2022 | - | 11 | 10 |  | SL |  | 10 | - |
| 07.04.2022 | 14 | 12 | - | P5 | SL | NyÅ | 17 | 15 (10) |
| 08.04.2022 | 15 | - | 11 | P6 |  |  | - | 21 (5) |
| 09.04.2022 | - | - | 12 |  |  |  | - | - |
| 10.04.2022 | 16 | 13 | 13 | P5, P6 | SL |  | 18 | 22 (21) |
| 11.04.2022 | 17 | - | - |  | SL |  | - | 7 (6) |
| 12.04.2022 | 18 | - | - |  |  |  | - | 17 (16) |

all three aircraft were well separated. HALO stayed mostly at 9-13 km altitude for remote sensing measurements, while Polar 5 also conducted remote sensing observations at 3 km altitude. To quantify the surface energy budget, Polar 5 measured in low altitude close to the surface during two flights. Polar 6 frequently changed altitude mostly in the atmospheric boundary layer between 0.1 km and 2 km, and regularly (once a flight) profiled the atmosphere up to 4 km altitude.

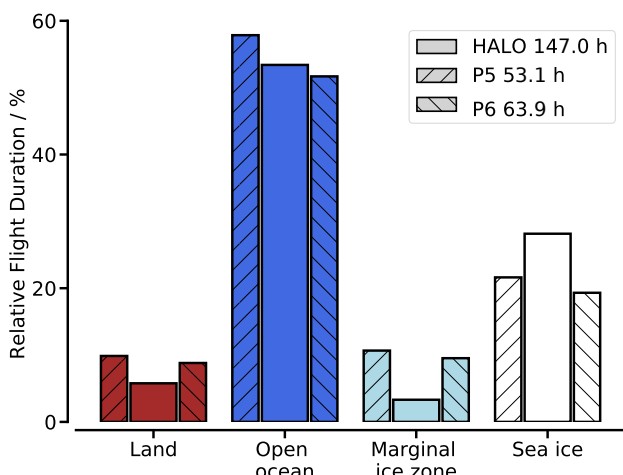

**Figure 2.** Classification of flight times over specific surface types separated for all aircraft individually.

## 2.3 Instrumentation

HALO–$(\mathcal{AC})^3$ made use of the experience in operating Polar 5, Polar 6, and HALO for atmospheric research and adapted the instrumentation from previous campaigns. HALO was set up for cloud and atmospheric remote sensing observations from high altitudes and in-situ sampling of basic atmospheric parameters by dropsondes. The instrumentation onboard of HALO is listed in Table 2 and is identical to the cloud observatory instrument package applied in a series of missions such as the Next Generation Remote Sensing for Validation Studies (NARVAL) mission and the Elucidating the role of cloud–circulation coupling in climate (EUREC$^4$A) campaign (Stevens et al., 2019; Konow et al., 2021).

On Polar 5 and Polar 6, separate payloads for remote sensing (Polar 5) and in-situ measurements (Polar 6) were installed in a similar configuration as operated during ACLOUD (Ehrlich et al., 2019). The instruments on board of Polar 5 and Polar 6 are listed in Table 3 including the main measured quantities and references providing detailed descriptions of the instruments and their uncertainties. Basic measurements of meteorological parameters and the radiative energy budget were conducted on both aircraft. The remote sensing (Polar 5) and cloud microphysical instruments (Polar 6) were operated almost identical setup during the AFLUX and MOSAiC-ACA campaigns (Mech et al., 2022a). For HALO–$(\mathcal{AC})^3$, the cloud microphysical devices was extended by a Cloud Probe with Polarization Detection (BCPD, Beswick et al., 2014). One major change of the Polar 6 instrumentation compared to Ehrlich et al. (2019) was in the aerosol particle microphysical instrumentation. The second ultra-high sensitivity aerosol spectrometer (UHSAS) was replaced by the High-volume flow aERosol particle filter sAmpler (HERA), a Mobility Particle Size Spectrometer (MPSS), and a miniature Cloud Condensation Nuclei Counter (mCCNC).

Table 2: Overview of the instrumentation of HALO and the measured quantities that are part of the data base. $\lambda$ is wavelength, $\nu$ is frequency, $T$ is air temperature, and $p$ is atmospheric pressure. RH is relative humidity, FOV is field of view.



| Instrument | Measured Quantities, Range, and Sampling Frequency | Reference |
|---|---|---|
| **Meteorology** | | |
| BAHAMAS | $T$, $p$, RH Wind Vector, 10 Hz | Krautstrunk and Giez (2012) |
| Dropsondes | Vertical Profile of $T$, $p$, RH, 2 Hz, | Vaisala (2020) |
| | Vertical Profile of Horizontal Wind Vectors, 4 Hz | |
| **Radiative Energy Budget** | | |
| BACARDI Pyranometer | Solar Irradiance (Upward, Downward, | Ehrlich et al. (2023) |
| | Broadband $\lambda = 0.2 - 3.6\,\mu m$), 10 Hz | |
| BACARDI Pyrgeometer | Terrestrial Irradiance (Up- and Downward, | Ehrlich et al. (2023) |
| | Broadband $\lambda = 4.5 - 42.0\,\mu m$), 10 Hz | |
| SMART | Spectral Irradiance (Downward $\lambda = 0.32 - 2.1\,\mu m$), 1 Hz | Wolf et al. (2019) |
| **Remote Sensing** | | |
| WALES | Backscatter Coefficient ($\lambda = 532\,nm$), 1 Hz | Wirth et al. (2009) |
| | Particle Linear Depolarization ($\lambda = 532\,nm$), 1 Hz | |
| | Water Vapor Molecular Density (at $\lambda = 935\,nm$), 12 s | |
| HAMP Active | Radar Reflectivity Factor, Doppler Spectra, $\nu = 35\,GHz$, 1 Hz | Mech et al. (2014) |
| HAMP Passive | Brightness Temperature, 25 Channels $\nu = 22.24 - 190.81\,GHz$, Nadir, 1 Hz | Mech et al. (2014) |
| specMACS | Spectral Radiance (Upward, Swath = $32.7° - 35.5°$, $\lambda = 0.4 - 2.5\,\mu m$), 30 Hz | Ewald et al. (2016) |
| | Polarized Radiance, RGB Color Channels, 2D fields 91° x 117°, 8 Hz | Weber et al. (2024) |
| VELOX | Spectral Brightness Temperature, 6 Channels $\lambda = 7.7 - 12\,\mu m$, | Schäfer et al. (2022) |
| | 2D Fields 35.5° x 28.7°, 635 x 507 pixels, 100 Hz | |
| KT-19 | Brightness Temperature (Upward nadir, $\lambda = 9.6 - 11.5\,\mu m$), 20 Hz | Schäfer et al. (2022) |

Table 3: Overview of the instrumentation of Polar 5 and Polar 6 and the measured quantities that are part of the data base. $\lambda$ is wavelength, $\nu$ is frequency, $T$ is air temperature, and $p$ is atmospheric pressure. RH is relative humidity, FOV is field of view, PNSD is the particle number size distribution, rBC refractory black carbon, SS is the level of supersaturation, and $D_{\mathrm{p}}$ symbolizes the particle diameter.

| Aircraft Reference | Instrument | Measured Quantities, Range, and Sampling Frequency | Reference |
|---|---|---|---|
| **Meteorology** | | | |
| P5&P6 | Nose Boom | $T$, $p$, RH, Wind Vector, 100 Hz | Hartmann et al. (2019) |
| | | | Tetzlaff et al. (2015) |
| P5 | Dropsondes | Vertical profile of $T$, $p$, RH, 2 Hz, | Vaisala (2020) |



| | | Vertical profile of Horizontal Wind Vector, 4Hz | |
|---|---|---|---|
| **Radiative Energy Budget** | | | |
| P5&P6 | Pyranometer | Solar Irradiance (Up- and Downward, | Ehrlich and Wendisch (2015) |
| | | Broadband $\lambda = 0.2 - 3.6\,\mu\text{m}$), 20 Hz | Philipona et al. (1995) |
| P5&P6 | Pyrgeometer | Terrestrial Irradiance (Up- and Downward, | Ehrlich and Wendisch (2015) |
| | | Broadband $\lambda = 4.5 - 42.0\,\mu\text{m}$), 20 Hz | Philipona et al. (1995) |
| P5 | SMART | Spectral Irradiance (Up- and Downward, $\lambda = 0.4 - 1.8\,\mu\text{m}$), 2 Hz | Bierwirth et al. (2009) |
| | | Spectral Radiance (Upward, FOV = 2.1°, $\lambda = 0.4 - 1.0\,\mu\text{m}$), 2 Hz | |
| **Remote Sensing** | | | |
| P5 | AISA Eagle/Hawk | Spectral Radiance (Upward, Swath = 36°, $\lambda = 0.4 - 2.5\,\mu\text{m}$), 20-30 Hz | Schäfer et al. (2013) |
| P5&P6 | Fish-Eye Camera | Spectral Radiance (Lower Hemisphere, RGB Channels), 4-6 s | Carlsen et al. (2020) |
| P5 | AMALi | Particle Backscattering Coefficient ($\lambda = 355, 532$ nm), Cloud Top Height, | Stachlewska et al. (2010) |
| | | Particle Depolarization ($\lambda = 532$ nm), 1 Hz | |
| P5 | MiRAC-A | Radar Reflectivity Factor, Doppler Spectra, $\nu = 94$ GHz, tilted by 25°, 1 Hz | Küchler et al. (2017) |
| | | Brightness Temperature, $\nu = 89$ GHz, tilted by 25°, 1 Hz | Mech et al. (2019) |
| P5 | HATPRO | Brightness Temperature, $\nu = 7 \times 22.24 - 31.4$ and | Rose et al. (2005) |
| | | $\nu = 7 \times 51.26 - 58.00$ GHz, Upward nadir, 1-2 Hz | |
| P5&P6 | KT-19 | Brightness Temperature (Upward nadir, $\lambda = 9.6 - 11.5\,\mu\text{m}$), 20 Hz | Schäfer et al. (2022) |
| **Cloud Microphysical Properties** | | | |
| P6 | 2D-S | Cloud PNSD, 2D Particle Images, $D_\text{p} = 10 - 1280\,\mu\text{m}$, 1 Hz | Lawson et al. (2006) |
| P6 | Polar Nephelometer | Non-normalized Volumetric Scattering Phase Function, 1 Hz | Gayet et al. (1997) |
| | | Asymmetry Parameter, Extinction Coefficient, 1 Hz | |
| P6 | CDP | Cloud PNSD, $D_\text{p} = 3 - 50\,\mu\text{m}$, 1 Hz | Lance et al. (2010) |
| P6 | CIP | Cloud PNSD, 2D Particle Images, $D_\text{p} = 15 - 960\,\mu\text{m}$, 1 Hz | Baumgardner et al. (2001) |
| P6 | PIP | Cloud PNSD, 2D Particle Images, $D_\text{p} = 100 - 6400\,\mu\text{m}$, 1 Hz | Baumgardner et al. (2001) |
| P6 | BCPD | Cloud PNSD, $D_\text{p} = 2 - 42\,\mu\text{m}$, Particle Shape, 1 Hz | Lucke et al. (2023) |
| P6 | Nevzorov Probe | LWC, TWC, 1 Hz | Korolev et al. (1998) |
| **Aerosol Microphysical Properties** | | | |
| P6 | CPC TSI-3010 | Aerosol and Cloud Residual Number Concentration, $D_\text{p} > 10$ nm, 1 Hz | Mertes et al. (1995) |
| P6 | UHSAS | Aerosol and Cloud Residual PNSD, $D_\text{p} = 65$ nm $- 1\,\mu\text{m}$, 1 Hz | Cai et al. (2008) |
| P6 | Grimm Sky-OPC | Aerosol PNSD, $D_\text{p} = 250$ nm $- 5\,\mu\text{m}$, 6 s | Bundke et al. (2015) |
| P6 | HERA | Aerosol Particle Filter Sampling, INP concentration, $2 - 140$ min | Grawe et al. (2023) |
| P6 | mCCNC | CCN Number Concentration, SS = 0.1 %, 1 Hz | Roberts and Nenes (2005) |
| P6 | MPSS | Aerosol PNSD, $D_\text{p} = 10$ nm $- 800$ nm, 5 min | Wiedensohler et al. (2012) |
| **Aerosol Chemistry** | | | |
| P6 | ALABAMA | Single Particle Composition (Refractory, Non-Refractory), | |
| | | $D_\text{va} = 230 - 3000$ nm ($D_{50}$), up to 10 Hz | Clemen et al. (2020) |





| P6 | PSAP | Particle Absorption Coefficient ($\lambda = 565\,\mathrm{nm}$), $\sigma = 0 - 5 \cdot 10^{-6}\,\mathrm{m}^{-1}$, $10\,\mathrm{s}$ | Bond et al. (1999) |
| | | | Mertes et al. (2004) |
| P6 | SP2 | rBC Mass/Number Concentration, PNSD, rBC Mass: $0.32 - 290\,\mathrm{fg}$, | Zanatta et al. (2023) |
| | | $D_\mathrm{p} = 70 - 675\,\mathrm{nm}$, Single Particle Data | |
| **Trace Gas Chemistry** | | | |
| P6 | Aerolaser AL5002 | CO-Concentration, $0 - 100{,}000\,\mathrm{ppbv}$, $1\,\mathrm{Hz}$ | Gerbig et al. (1999) |
| P6 | Licor 7200 | $CO_2$ Concentration, $0 - 3000\,\mathrm{ppmv}$, $1\,\mathrm{Hz}$ | Burba et al. (2010) |
| | | $H_2O$ Concentration, $0 - 60\,\mathrm{mmol\,mol}^{-1}$, $1\,\mathrm{Hz}$ | |
| P6 | 2BTech O3 Monitor | $O_3$-Concentration, $0 - 250\,\mathrm{ppmv}$, $2\,\mathrm{s}$ | Johnson et al. (2014) |
| P6 | WVSS-II | $H_2O$ Concentration, $50 - 40000\,\mathrm{ppmv}$, $1\,\mathrm{Hz}$ | Vance et al. (2015) |
| | | Cloud Water Content, $10 - 2000\,\mathrm{mg\,m}^{-3}$, $1\,\mathrm{Hz}$ | |

## 3 Datasets

All data are published in the World Data Center PANGAEA (Felden et al., 2023) as instrument-separated data subsets and are marked by the tag "HALO–$(\mathcal{AC})^3$". Three collections of data sets for HALO, Polar 5, and Polar 6 link the data subsets by aircraft for a more structured overview.

- HALO: (Ehrlich et al., 2024a, https://doi.org/10.1594/PANGAEA.968885)

- Polar 5: (Mech et al., 2024a, https://doi.org/10.1594/PANGAEA.968883)

- Polar 6: (Herber et al., 2024, https://doi.org/10.1594/PANGAEA.968884)

All data subsets, separated into the scientific subject, are introduced below. The subsection titles indicate, which aircraft the data sets refer to.

### 3.1 Measurements of basic aircraft and fundamental meteorological parameters

#### 3.1.1 HALO - Basic aircraft data

The basic aircraft data obtained from instruments mounted on HALO (wind vector, thermodynamic state of the ambient air) were measured with the Basic HALO Measurement and Data System (BAHAMAS, Giez et al., 2021; Krautstrunk and Giez, 2012). For HALO–$(\mathcal{AC})^3$, all processed data are published with $10\,\mathrm{Hz}$ sampling frequency in the HALO data base (Giez et al., 2022, https://halo-db.pa.op.dlr.de/mission/130). The measurements include aircraft location, speed and attitude, the three-dimensional (3D) wind vector, air pressure, static air temperature, and humidity. All data were corrected for dynamical effects due to the aircraft movement. The calibration procedure of the wind vector measurements is documented by Giez et al. (2021), the static pressure measurements by Giez et al. (2020). A full uncertainty analysis of all measured quantities in high latitude conditions is provided by Giez et al. (2023). The humidity measurements of BAHAMAS were evaluated in an inter-comparison





published by Kaufmann et al. (2018). For users only interested in the position data of HALO, master tracks of longitude, latitude, and altitude derived from the global positioning system (GPS) are published in PANGAEA (Ehrlich et al., 2024b).

### 3.1.2 Polar 5 / Polar 6 - Basic aircraft data and high frequency nose boom data

Slow and high frequency measurements of wind, air temperature, and humidity were obtained on Polar 5 and Polar 6 from two

identical nose boom systems and sensors mounted in a Rosemount housing as described by Mech et al. (2022a). The processed datasets for the nose boom measurements are available in ASCII format on PANGAEA in a 100 Hz resolution (Lüpkes et al., 2024a) and in 1 Hz resolution (Lüpkes et al., 2024b). For users only interested in the position data of the aircraft, master tracks of GPS longitude, latitude, and altitude are published for Polar 5 and Polar 6 (Herber et al., 2022a, b).

The 3D wind vector was derived from Aventech (Aventech Research Inc., Canada) five-hole probes for high frequency

pressure measurements combined with accurate position data, which were obtained from a high precision GPS and from an inertial navigation system (INS). Pitch and roll angles were delivered at an accuracy of $0.1°$ while true heading angles have an accuracy of $0.4°$. Finally, horizontal wind components were obtained with an absolute accuracy of $0.2\,\mathrm{m\,s^{-1}}$ for straight and level flight sections. Vertical wind was obtained only as the deviation from the average vertical wind. For sections of several kilometers length, an accuracy of about $0.05\,\mathrm{m\,s^{-1}}$ of the vertical wind speed relative to the average wind is estimated.

Temperature was measured after correcting the adiabatic heating effect of the air by the dynamic pressure at an absolute accuracy of $0.3\,\mathrm{K}$ and with a resolution of $0.05\,\mathrm{K}$.

For slow air humidity measurements (frequency of 1 Hz), a Vaisala HMT-333 with a temperature and HUMICAP humidity sensor was mounted in a Rosemount housing. Based on the temperature measurements (uncertainty of $0.1\,\mathrm{K}$), the humidity data were corrected for adiabatic heating and reach an accuracy of $2\,\%$. All data were recorded with a frequency of 100 Hz.

Note that the calibration of the 100 Hz data are only valid for straight and level flights, when using these for the calculation of turbulent fluxes. The latter are not provided in the dataset. Users who want to calculate fluxes with the eddy covariance method should consider only such flight sections where the aircraft was flying in a straight line.

During the HALO–$(\mathcal{AC})^3$ campaign the sensors mounted at the nose boom showed icing problems. As a result, high resolution data are not available from all flights. Problems concerned especially data from Polar 5, where we cannot provide wind

data from nine flights while from Polar 6 wind data are missing from five flights. Icing is clearly marked in the published data.

### 3.1.3 HALO / Polar 5 - Dropsonde data

Dropsondes were released from HALO and Polar 5 to measure vertical profiles of atmospheric air pressure, temperature, relative humidity, and the horizontal wind vector. HALO exclusively used the Vaisala RD41 dropsondes (Vaisala, 2020), whereas

the Polar 5 used the Vaisala RD94 for nine launches and Vaisala RD41 for the rest. Our data processing does not distinguish between the two instrument models. On both aircraft, the Advanced Vertical Atmospheric Profiling System (AVAPS) was operated for the release and data acquisition of the dropsondes. The spatial coverage of the dropsondes released dur-



ing HALO–$(\mathcal{AC})^3$ is mostly well distributed along the flight tracks (Fig. 3). However, due to limitations in flight operations, dropsondes were less frequently released in Danish airspace west of $0°$ longitude.

The dropsonde data are provided (George et al., 2024) in the same format as the Joint dropsonde Observations of the Atmosphere in tropical North atlaNtic meso-scale Environments (JOANNE) data set from the EUREC⁴A field campaign (George et al., 2021) and maintain the same definitions of the data-processing levels from Level-0 to Level-3 as in JOANNE. For processing the dropsondes' raw data (Level-0), the Atmospheric Sounding Processing ENvironment (ASPEN Version 4.0.0, Martin and Suhr, 2024) was used and provides the basic quality controlled data (Level-1). All dropsondes underwent

a pre-flight reconditioning to restore the humidity sensors calibration status from drift due to trace-gas pollutants, as also explained in Vömel et al. (2021). The dry bias correction applied in JOANNE was therefore omitted.

Level-2 data includes sondes that passed an additional quality control (successful launch and landing detection) and only contain variables that were measured. Both Level-1 and Level-2 data are provided as one file per sounding. Level-3 provides a single NetCDF file, which combines all valid Level-2 soundings from HALO and P5 in a uniform vertical grid-spacing

of 10 m. We exclude any Level-4 products here (otherwise present in JOANNE), because of varying sampling strategies of the campaign, which restricts a simplified statistical analyses of area-averaged properties such as by George et al. (2023). Nevertheless, the retrieval for estimating the area-averaged properties during HALO–$(\mathcal{AC})^3$ are detailed in the study by Paulus et al. (2024) and the user is recommended to use the provided Level-3 data for following their estimation.

### 3.1.4  HALO - Communicating dropsonde data to GTS and assimilation to ECMWF services

Observations by the dropsondes released from HALO were submitted live during the flights via satellite communication to the Global Telecommunication System (GTS) to make the measurements available for data assimilation. On basis of the European Centre for Medium-Range Weather Forecasts' (ECMWF) Integrated Forecasting System (IFS) the assimilated data are evaluated here. Data were sent only for a set of mandatory (main pressure levels) and significant levels (local extremes). Thinning and quality control within data assimilation did further reduce data coverage. In total, data of 216 dropsondes were assimilated

for 14 analysis times. The number of sondes for each analysis time varies between 4 and 41 sondes. The geographical location of the assimilated data are illustrated in Fig. 3a. There is no tendency to prefer data in sparsely covered high latitudes. Also data from high spatially resolved flight patterns such as from RF12 on 1 April 2022 were assimilated (41 sondes).

The vertical distribution of data availability in the native vertical grid of IFS is shown in Fig. 3b-d individually for wind speed, air temperature, and humidity. The distribution indicates an increased data coverage for the mandatory pressure levels.

No air humidity measurements were used in the stratosphere where the accuracy of measurements is large. In the troposphere, the number of assimilated air humidity data is similar to the amount of assimilated air temperature data. In total, 19,131 observations from 216 sondes were assimilated, which distributes to 5,867 temperature observations (from 216 sondes), 4,760 wind observations (from 216 sondes), and 3,744 humidity observations (from 214 sondes).

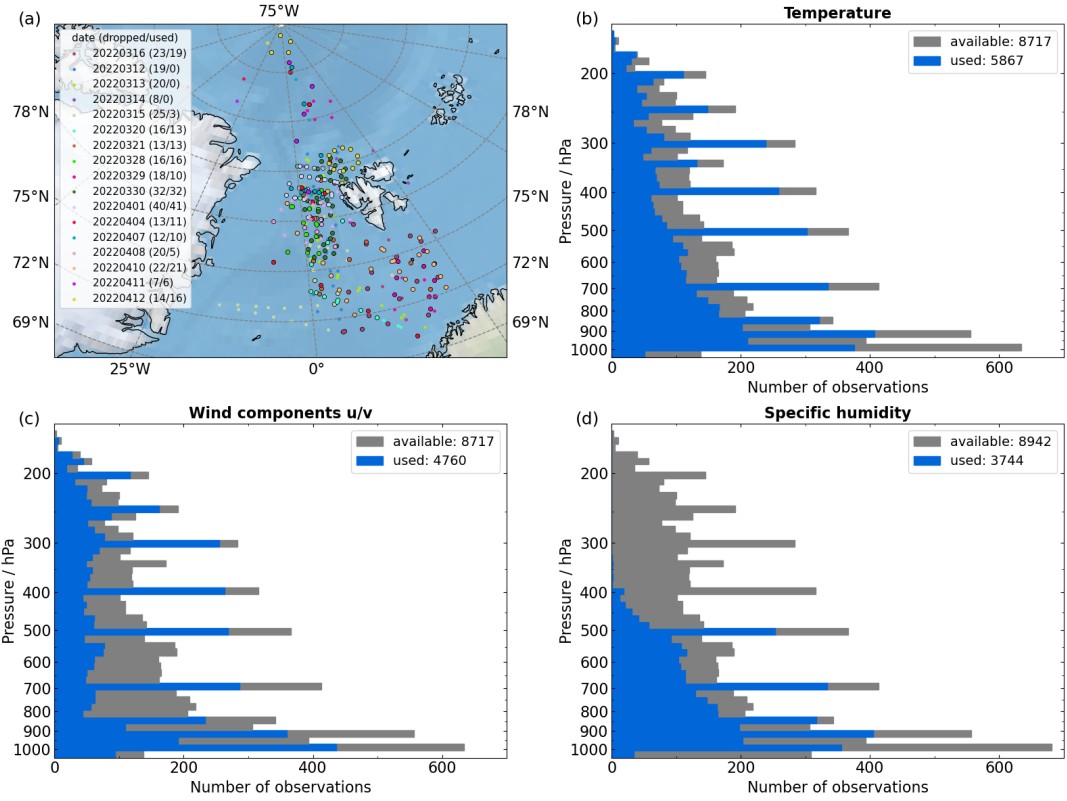

**Figure 3.** Geographic locations (a) and altitudes (b-d) of dropsonde measurements of air temperature (b), horizontal wind speed (c) and specific humidity (d), which are available in the HALO–$(\mathcal{AC})^3$ data set and which were assimilated into GTS.

### 3.1.5 HALO - Quasi-Lagrangian matches

The HALO–$(\mathcal{AC})^3$ campaign realized a quasi-Lagrangian flight strategy. As described by Wendisch et al. (2024), air masses were sampled by HALO twice or even more often between two research flights or during one flight. To promote the analysis of the air mass evolution, the locations of these quasi-Lagrangian matches are published in PANGAEA. Kirbus et al. (2024a) summarizes the matches that occurred within one flight day, while matches between two consecutive flight days are listed by Kirbus et al. (2024b). The data sets include only matches with a minimum threshold of one hour between the first and second sampling.

The matches were derived from trajectory calculations based on the wind fields from the ECMWF reanalysis version 5 (ERA5, Hersbach et al., 2020) and the Lagrangian Analysis Tool (LAGRANTO, Sprenger and Wernli, 2015). ERA5 data has a native temporal resolution of one hour and horizontal resolution of 31 km. For each research flight of HALO, air mass trajectories were initialized in one minute resolution along the flight track via a bi-linear interpolation of the hourly ERA5 fields. To account for the ERA5 temporal and spatial resolution, 30 trajectories were distributed in a circle of 30 km radius around





HALO. Vertically, trajectories were initialized in all altitudes below HALO with a 5 hPa resolution. A quasi-Lagrangian match was registered, when any of the initialized air masses crossed the flight track of HALO at a later time within 30 km distance. Details on this approach and statistics on the number of quasi-Lagrangian matches are given by Wendisch et al. (2024). An
uncertainty analysis is presented by Kirbus et al. (2024). Using different reanalysis data, they showed that the input wind field does not significantly change the trajectories. Dropsonde data, including wind speed and direction, were assimilated by ECMWF and, therefore, were considered in the ERA5 data. This can be assumed to significantly improve the reliability of the trajectory calculations and quasi-Lagrangian matches.

## 3.2   Radiative energy budget measurements

### 3.2.1   HALO / Polar 5 / Polar 6 - Broadband irradiances

On HALO, broadband solar and thermal-infrared irradiances were measured with the Broadband AirCrAft RaDiometer Instrumentation (BACARDI, Ehrlich et al., 2023) consisting of CMP22 pyranometers and CGR4 pyrgeometers manufactured by Kipp&Zonen. A similar setup of identical radiometers was used on Polar 5 and Polar 6 (Ehrlich and Wendisch, 2015; Ehrlich et al., 2019). Detailed specifications, data processing and corrections, and the instrument performance of BACARDI onboard
HALO are documented in Ehrlich et al. (2023). The calibrated and quality-checked broadband irradiances (upward/downward and solar/thermal-infrared) from HALO and Polar 5/Polar 6 are provided in separate data sets (Luebke et al., 2023; Becker et al., 2023).

The radiometer performance on Polar 5 and Polar 6 was evaluated by Ehrlich et al. (2019). Measurements during HALO–$(\mathcal{AC})^3$ are characterized by cold environments (sensor temperatures below $-55\,°C$) and a dominance of large solar zenith
angles (SZAs). For these conditions, the calibration uncertainties increase (Su et al., 2008). On the other hand, due to the low incoming solar irradiance in Arctic conditions, the solar leakage effect of the pyrgeometer effecting the downward thermal-infrared irradiance is low (Ehrlich et al., 2023). In total, the uncertainties of the irradiances reach values in the order of up to $10\,\mathrm{W\,m^{-2}}$.

The published BACARDI data have a sampling frequency of $10\,\mathrm{Hz}$ and include all corrections described by Ehrlich et al.
(2023). Onboard Polar 5 and Polar 6, the broadband irradiances were recorded at a frequency of $20\,\mathrm{Hz}$. Due to lower ascent and descent rates of Polar 5 and Polar 6, the dynamic thermal offset needed not be corrected for in the data set. Instead, a basic temperature change flag was added to the data set, which can be used to exclude the irradiances where the thermal offset becomes relevant. The flag identifies temperature gradients exceeding $0.5\,\mathrm{K\,min^{-1}}$ (smoothed with a one-minute rectangular filter). The attitude correction of downward solar irradiance can only be applied for the direct solar radiation, e.g., when no
clouds were present above the aircraft. Therefore, the final data include two versions of the downward solar irradiances, an uncorrected one for cloudy conditions and a corrected product to be used in cloud-free conditions. To distinguish between these two scenarios, a cloud flag, derived from the downward solar irradiance and upward-looking camera imagery, is provided in the published data sets. Due to frequent passages through super-cooled clouds by Polar 5 and Polar 6, probable icing is marked by an icing flag as described by Ehrlich et al. (2019). In both data sets, simulated cloud-free downward solar irradiances are





included. The radiative transfer simulations were set up as documented in Ehrlich et al. (2023), assuming a bright sea ice surface. The simulations were used for cloud detection, but are of limited use for calculations of the cloud radiative effects, where changes of the surface albedo need to be considered (Stapf et al., 2020).

### 3.2.2 HALO / Polar 5 - Spectral solar irradiance

Spectrally resolved downward (Polar 5, HALO) and upward (Polar 5 only) solar irradiance were measured using two versions of the Spectral Modular Airborne Radiation measurement sysTem (SMART, Wendisch et al., 2001; Wolf et al., 2019). The data are published separately for HALO (Röttenbacher et al., 2023) and Polar 5 (Jäkel et al., 2023a).

SMART was radiometrically calibrated before, after, and during the campaign following Wolf et al. (2019). The cosine response of the optical inlets was characterized with laboratory calibrations, especially for the large values of SZA commonly encountered during HALO–$(\mathcal{AC})^3$. This calibration was applied in the data processing accounting for the amount of direct solar radiation in cloudless conditions. The processed data of the two types of grating spectrometers were merged and interpolated to $1\,\mathrm{nm}$ resolution in the visible and near infrared spectral range (wavelength below $1\,\mathrm{\mu m}$) and to $5\,\mathrm{nm}$ resolution in the shortwave infrared spectral range (wavelength above $1\,\mathrm{\mu m}$). Due to the lower sensitivity and the stronger noise at the edges of the spectrometers, the HALO data are limited to wavelength between 0.32 to $2.1\,\mathrm{\mu m}$, and the Polar 5 data to 0.4 to $1.8\,\mathrm{\mu m}$ wavelength. Polar 5 data are provided with $2\,\mathrm{Hz}$ frequency, while HALO measurements required a longer sampling time and are published with $1\,\mathrm{Hz}$ temporal resolution.

On HALO, the optical inlet mounted on top of the fuselage was actively stabilized for all flights apart from RF18 where the stabilization was fixed and aligned with the fuselage. For this flight, the downward irradiance was corrected for the aircraft movement following Bannehr and Schwiesow (1993) and Boers et al. (1998). On Polar 5, all optical inlets were actively stabilized (Wendisch et al., 2001). To minimize the remaining instrumental uncertainties, the Polar 5 data set was filtered for large SZA (less than $82°$) and aircraft pitch and roll angle ($\pm 4.5°$). HALO data are not filtered but should be used with caution for these conditions. For large SZAs around $80°$, as was the case with HALO–$(\mathcal{AC})^3$, the uncertainty of the measured irradiance at flight level is increased compared to observations at smaller SZAs. The known uncertainty components (radiometric calibration, cosine correction, sensor tilt) of SMART were re-evaluated with regard to specific conditions during HALO–$(\mathcal{AC})^3$. The uncertainty of the downward and upward irradiances adds up to $6.5\,\%$ and $6.0\,\%$, respectively.

## 3.3 Remote sensing observations

### 3.3.1 HALO / Polar 5 - Lidar

Polar 5 and HALO carried lidar systems for cloud, aerosol, and water vapor remote sensing. On HALO, backscatter lidar and water vapor differential absorption lidar profiles were recorded using the airborne demonstrator for the WAter vapor Lidar Experiment in Space (WALES, Wirth et al., 2009). The data set is published by Wirth and Groß (2024) in netCDF format and provides time series of profiles of backscatter ratio, particle linear depolarization depolarization, and water vapor molecular density measured along the flight path of HALO on 17 days. Backscatter ratio and aerosol depolarization data were





derived with one second time resolution and 15 meter vertical resolution at a wavelength of 532 nm. The backscatter profiles are extinction-corrected using the High Spectral Resolution Lidar (HSRL) method (Esselborn et al., 2008). The water vapor profiles have a time resolution of 12 s and a vertical binning of 15 m. For $H_2O$, the vertical resolution is lower than given by the binning, where the real resolution is determined by an averaging kernel, which is constant over height and has a full width at half maximum (FWHM) of 250 m. All data were regridded to a constant altitude scale over mean sea level, irrespective of the actual flight altitude. The data files include a flag field build from seven different bits that indicate the possible problems with each individual data point. Details on the flag field and how to interpret the problems are described in the netCDF attributes. The data are considered of good quality if no flag is set, i.e., if the flag is zero. Note that for a non-zero flag, the data value itself is not replaced by a fill-value.

On Polar 5, the Airborne Mobile Aerosol Lidar (AMALi, Stachlewska et al., 2010) was operated to derive cloud mask and cloud top height that are published by Mech et al. (2024d). The measured profiles of backscattered intensities at 355 nm and 532 nm wavelength were processed similarly to previous campaigns (Ehrlich et al., 2019; Mech et al., 2022a). This processing includes corrections for the background signal, range, incomplete overlap, and drift of the so-called base line. The corrected backscatter intensities were gridded and used to calculate the logarithmic volume attenuated backscatter coefficients. The cloud detection only applied the parallel-polarized attenuated backscatter coefficient at 532 nm wavelength. Cloud top height was determined from the backscatter gradient and magnitude and provided with a vertical resolution of 7.5 m. The published data contain the cloud mask and cloud top height in 1 s resolution.

### 3.3.2 HALO - Radar and microwave radiometer

Active and passive microwave remote sensing observations were performed on HALO using the HALO Microwave Package (HAMP, Mech et al., 2014). HAMP combines a 35 GHz cloud and precipitation radar and passive microwave radiometer measuring at 26 frequency channels between 20 GHz and 183 GHz. The data of both components are published in a unified format by Dorff et al. (2023) and include merged time-series of brightness temperatures from the radiometers, radar reflectivity factor, and radar linear depolarization ratio. This unification is based on Konow et al. (2019) and synchronizes the measurements from both devices into a collocated temporal 1 Hz resolution applicable for joint analysis. The vertically resolved radar measurements were further projected on a joint vertical grid of 30 m resolution.

In the unified data set, radar reflectivity factors are offset-calibrated using the method described by Ewald et al. (2019). For the radiometers, imperfect pre-flight calibration can cause an offset in the measured brightness temperatures. Therefore, cloud-free dropsonde measurements of air temperature and humidity were used to post-calibrate following the procedure proposed by Jacob (2020). The atmospheric profiles measured in cloud-free conditions were used as input for forward simulations using the Passive and Active Microwave radiative TRAnsfer tool (PAMTRA, Mech et al., 2020), which were then compared to the observation to determine potential offsets. For both measurement devices, the post-calibrated data were quality checked (meaning cleaned of outliers and gap-filled). Radar data obtained during turns of HALO were removed due to disturbances related to side lobes. An adherent surface mask distinguishes between three predominant over-passed surface types (land, sea, and sea-ice cover), considering the satellite-based AMSR2 sea ice cover (Spreen et al., 2008).



### 3.3.3 Polar 5 - Radar and microwave radiometer

On Polar 5, equivalent radar reflectivities at 94 GHz frequency were measured with a frequency-modulated continuous wave radar, the active component of the Microwave Radar/radiometer for Arctic Clouds (MiRAC-A, Mech et al., 2019). Additionally, MiRAC-A provides brightness temperatures from a passive channel at 89 GHz frequency. Radar reflectivities and brightness temperatures are published in a combined data set with a temporal resolution of 1 Hz by Mech et al. (2024b). MiRAC-A is mounted on Polar 5 with an inclination of 25° backward with respect to the nadir and aircraft flight directions. In post-processing, the radar reflectivities were corrected to nadir and resampled on a 5 m vertical grid. Published brightness temperatures from the 89 GHz channel have not been corrected for the viewing geometry and are still along the inclined observation path. Details on the instruments' resolution, accuracy, and processing are described by Mech et al. (2022a).

During HALO–$(\mathcal{AC})^3$, the Humidity And Temperature PROfiler (HATPRO, Rose et al., 2005) was operated onboard Polar 5. HATPRO provides brightness temperatures in 14 channels, seven vertically polarized at the water vapor absorption line at 22.24 GHz (K-band) and seven horizontally polarized around the oxygen absorption complex at 60 GHz (V-band). All data are published with a temporal resolution of approximately 1 Hz by Mech et al. (2024c). The data were post-processed identically to previous campaigns such as MOSAiC-ACA (Mech et al., 2022a).

### 3.3.4 HALO / Polar 5 - Solar spectral and polarized imaging

Partly identical, spectral push-broom imagers for the solar spectral range were operated on HALO and Polar 5. The spectrometer of the Munich Aerosol Cloud Scanner (specMACS, Ewald et al., 2016; Weber et al., 2024) on HALO and the Airborne Imaging Spectrometer for Applications (AISA) Eagle/Hawk system (Ehrlich et al., 2019; Ruiz-Donoso et al., 2020; Klingebiel et al., 2023) on Polar 5 measured two-dimensional (2D) fields of reflected spectral radiances in a spectral range between 400 nm and 2500 nm. Having a swath with a similar field of view (around 35°) and flying in different altitude, the two systems provide different detail of the observed scenes down to 1 m at cloud top. Data are published separately for the two systems and are split into their single sensors.

On HALO, the spectral cameras of specMACS were operated in a nadir-looking configuration during HALO–$(\mathcal{AC})^3$. The data are published by Weber et al. (2024b) and provide radiance fields sampled with 30 Hz frequency. The processing of the raw data into spectral radiances, including attitude correction, radiometric and spectral calibration, was performed as described by Ewald et al. (2016). Due to technical problems with one of the spectrometers, the published data are limited to the spectral range from 1000 nm to 2400 nm.

On Polar 5, spectral radiances in the wavelength range from 400 nm to 993 nm were measured with the AISA Eagle instrument. Data cubes of the 1024 spatial across-track pixels and 504 spectral channels are published by Klingebiel et al. (2024) with the native frame rate of 20 Hz. Data processing and data evaluation was similar to the procedures presented in Ehrlich et al. (2019). Spectral radiances in the shortwave infrared wavelength range above 1000 nm wavelength, as provided by specMACS on HALO, were measured on Polar 5 by the AISA Hawk instrument for 384 across-track pixels and 288 spectral channels. Un-





fortunately, during HALO–$(\mathcal{AC})^3$ the instrument encountered significant challenges due to heavy condensation of main optical components. Post processing did not reliably remove these features and renders the collected data unsuitable for publication.

In addition to the spectral imagers, specMACS on HALO includes an additional polarized imaging system consisting of two 2D RGB polarization-resolving cameras with a maximum combined field of view of $91°\times117°$ measuring with 8 Hz temporal resolution (Weber et al., 2024). The data are published by Weber et al. (2024a) separately for each camera in form of mpeg videos combining the RGB images and images of the degree of linear polarization with the temporal resolution reduced by a factor of 10. The videos are useful for visualizing the general atmospheric and surface condition below HALO. Absolute

calibrated Stokes vectors rotated into the scattering plane can be computed from the polarization measurements as described by Weber et al. (2024) and can be made available upon request.

### 3.3.5   HALO - Thermal infrared spectral imaging

On HALO, thermal infrared remote sensing observations were performed by the Video airbornE Longwave Observations within siX channels (VELOX, Schäfer et al., 2022) system, which includes an actively cooled thermal-infrared imager. 2D

cloud-top and surface brightness temperature (BT) fields were derived from the observations and published by Schäfer et al. (2023).

The data comprise VELOX brightness temperature measurements for four narrow-band channels (central wavelengths and half-widths at $8.65\pm0.55\,\mu m$ (BT2), $10.74\pm0.39\,\mu m$ (BT3), $11.66\pm0.81\,\mu m$ (BT5), and $12.00\pm0.50\,\mu m$ (BT6)) and one broadband channel (7.7 µm to 12.0 µm). The measurements were processed according to Schäfer et al. (2022), including a destriping

procedure, an image filtering, and a pixel correction. Although VELOX operates a filter wheel and image sensor with a 100 Hz frame rate (one spectral filter per frame), the subsequent images are slightly shifted due to the aircraft movement. This shift was corrected to allow merging the measurements at different channels. These corrections reduced the image size to 635 x 507 pixels. To have a manageable data amount and file size, but remaining sufficient overlap of two subsequent images, the data are provided with a temporal resolution of 1 Hz. For the cold surface and atmosphere prevailing during HALO–$(\mathcal{AC})^3$,

the relative uncertainties of the low brightness temperatures are slightly higher compared to the tropical condition reported by Schäfer et al. (2022).

### 3.3.6   HALO / Polar 5 - Thermal infrared radiometer

On HALO and Polar 5, thermal infrared radiance in nadir direction (field of view of $2.3°$) were measured by KT19 (model KT19.85 II) infrared pyrometers. On HALO, the radiometer is integrated in the VELOX system (Schäfer et al., 2022), however,

data are published separately to provide the KT19 data with the full sampling frequency of 20 Hz (Schäfer et al., 2024). The KT19 measurements on Polar 5 are implemented in the data set of the broadband radiometer (Becker et al., 2023), which also were sampled with 20 Hz. Data are provided as brightness temperatures corresponding to the spectral range of the radiometer covering a narrow wavelength band between 9.6 µm and 11.5 µm. For HALO, a correction for the window transmissivity similar to the data processing of VELOX was applied.



### 3.3.7 Polar 5 / Polar 6 - Fish-eye camera

On Polar 5 and Polar 6, digital RGB cameras (Nikon D5) equipped with a downward-looking 180° fisheye lens measured the directional distribution of upward radiance of the entire lower hemisphere every 4-6 s. Data are published for the three spectral bands (red, green, and blue) by Jäkel and Wendisch (2024). A surface type classification based on the Polar 5 camera data are published by Jäkel et al. (2023b) providing the fraction of open water, sea ice, and melt ponds for each image.

The RGB images were recorded in raw data format. The post processing of the images included the rectification of the images with respect to flight attitude data and the transformation of the raw signal of each sensor pixel into calibrated radiances (Carlsen et al., 2020; Mech et al., 2022a). The final data set is re-binned to an angular-resolved radiance fields with $0.2°$ resolution. For data quality checks the nadir radiances were compared to the SMART radiance data. Following Carlsen et al. (2020), the measurement uncertainties are estimated at 4.5 %.

## 3.4 Cloud microphysical in-situ observations

To measure the microphysical properties of clouds, an advanced configuration of in-situ cloud probes was integrated on the wings and fuselage of the Polar 6 aircraft. The instruments are classified below depending on their operating principles as scattering instruments or optical array probes (OAPs). Additionally, a Nevzorov bulk probe was installed on the nose boom for cloud liquid water content (LWC) and total water content (TWC) measurements. All in-situ cloud probes, except for the Back-scatter Cloud Probe with Polarization Detection (BCPD), were deployed during previous aircraft Arctic field campaigns onboard of Polar 5. The instruments and corresponding data processing used here are identical to those used for the AFLUX and MOSAiC-ACA campaigns, and are described in more detail including uncertainties by Moser et al. (2023) and Mech et al. (2022a).

Data are published individually for the different cloud probes (see below) and in a combined version by Moser et al. (2023). The combined data make use of the different particle size ranges covered by the individual instruments and provide continuous particle number size distributions from 3–6400 μm and derived total particle number concentration, total cloud water, ice and liquid water content, total effective diameter ($D_{eff}$) and median diameter.

### 3.4.1 Polar 6 - Scattering cloud probes

Smallest cloud particles were counted and sized by the Cloud Droplet Probe (CDP, Lance et al., 2010) based on the intensity of forward scattered laser light (658 μm). The measurements were processed according to Moser et al. (2023) and published by Moser et al. (2023) with 1 Hz frequency. The data include primary measured particle number size distributions (PNSD) from 2.8 μm to 50 μm and derived quantities such as the total cloud particle number concentration (N), the effective diameter ($D_{eff}$), and the liquid water content (LWC). Similar to Kirschler et al. (2023), the calculations did neglect the presence of ice crystals in the size regime of the CDP, which is a reasonable assumption for Arctic low level clouds (McFarquhar et al., 2017; Korolev et al., 2017).





The BCPD measured similar microphysical properties as derived from the CDP, but with additional depolarization signal to identify the thermodynamic phase of the cloud particles (Beswick et al., 2014). The PNSDs were obtained in a slightly smaller size range from 2 - 42 µm. Measured PNSD and derived microphysical quantities are published by Lucke et al. (2023) with 1 Hz frequency. The thermodynamic phase of the cloud particles is provided as the number of ice crystals observed within 1 s

sampling time. Data processing and an assessment of differences between CPD and BCPD measurements are published by Lucke et al. (2023). The BCPD was integrated into the fuselage of Polar 6. The proximity to the aircraft skin likely affected its measurements, in the sense that the observed PNSD are altered with respect to those in the clouds. It is therefore advised to use the PNSD from the CDP instead. Furthermore, shattering of ice particles on the fuselage artificially increases the measured ice particle concentration. Currently, the BCPD should therefore only be used to assess the cloud phase (i.e., differentiate between

pure liquid water clouds, mixed-phase clouds and entirely glaciated clouds). A detailed discussion of the evaluation procedure is provided by Lucke et al. (2023).

The Polarnephelometer (PN) directly measures the non-normalized volumetric scattering phase function (i.e., angular scattering coefficients, ASCs) for cloud particles with diameters from a few micrometers to 1 mm using a collimated laser with a wavelength of 0.8 µm and a circular array of photodiodes (Gayet et al., 1997). The data are published in 1 Hz temporal resolu-

440 tion by Dupuy et al. (2024) and include the phase function for all 56 scattering angles. Additionally, extinction coefficient and the asymmetry parameter retrieved from the ASCs are provided to distinguish spherical from non-spherical cloud particles, as well as the dominant cloud thermodynamic phase.

### 3.4.2 Polar 6 - Optical array probes

Optical array probes (OAPs) recording shadow-graphs of droplet and ice particles were used to derive PNSD, N, $D_{eff}$ and

445 ice water content (IWC) from the 2D images (Baumgardner et al., 2001; Lawson et al., 2006; O'Shea et al., 2021; De La Torre Castro et al., 2023). Three OAPs were installed under the wing of Polar 6, the Cloud Imaging Probe (CIP), the Precipitation Imaging Probe (PIP), and the 2D Stereo Imaging Probe (2D-S). The OAPs differ in pixel quantity and resolution (64 diode array with 15 µm resolution for CIP, 64 diode array with 103 µm resolution for PIP, and 128 diode array with 10 µm resolution for 2D-S), which determine their observable particle size ranges (see Table 3). CIP and PIP data are included in

data set published by Moser et al. (2023). The 2D-S data are published by Dupuy et al. (2024) and are provided separately for the horizontal and vertical viewing direction. Due to the complexity of deriving microphysical properties from OAPs different solutions are included in the data sets, which can be selected depending on the focus of analysis.

### 3.4.3 Polar 6 - Nevzorov probe

The Nevzorov probe is a constant-temperature, hot-wire instrument designed for the airborne bulk measurements of LWC and

455 TWC (McFarquhar et al., 2017). Both quantities were corrected for the collision efficiency and published with 1 Hz temporal resolution by Lucke et al. (2024). The operation of the Nevzorov probe in mixed-phase clouds is challenging and requires an accurate estimation of the collection efficiencies. Laboratory measurements were performed to estimate the collection efficiencies in super-cooled cloud conditions (Lucke et al., 2022). For HALO–$(\mathcal{AC})^3$, the LWC and TWC were additionally corrected





by an iterative approach using the particle phase measurements by the BCPD (Lucke et al., 2023). Due to a malfunction of the Nevzorov probe data acquisition, data are available only for RF8–RF13.

### 3.5 Aerosol microphysical and chemical in-situ measurements

#### 3.5.1 Polar 6 - Particle inlets

To sample aerosol particles and cloud particle residuals (CPRs) using instrumentation inside the cabin of Polar 6, a standard aerosol inlet and a counterflow virtual impactor (CVI) inlet (Ogren et al., 1985) were deployed aboard Polar 6. Both inlets were applied in the same way as during ACLOUD (Ehrlich et al., 2019). A set of instruments, listed below, was connected to these inlets.

During HALO–$(\mathcal{AC})^3$, the number of super-cooled droplets exceeded by far the number of ice particles in mixed-phase clouds, i.e., the number of sampled CPRs were dominated by cloud droplet residuals. Due to the CVI operation principle (Mertes et al., 2005), the sampled CPRs are enriched with respect to ambient conditions. The respective enrichment factor (EF) needs to be applied to every instrument sampling CPRs and is therefore provided in the CVI data sets (Mertes and Wetzel, 2023). Moreover, two different inlet efficiencies must be considered. First, the sampling efficiency accounts for particles that were not counted due to the lower cut-off size of the CVI inlet. For the CVI on Polar 6, the cut-off size is around 10 μm due to the low flight velocity and implies that not all cloud droplets were collected. Second, a fraction of droplets was not sampled due to the substantial offset between droplet motion and aircraft heading. This effect is quantified by the aspiration efficiency for the droplet collection above the lower CVI cut-off size of 10 μm. Both efficiencies can be deduced by the comparison of the CPR and the size resolved droplet number concentration, where the latter can be derived from the in-situ cloud probes aboard Polar 6 (Ehrlich et al., 2019).

The standard aerosol inlet samples approximately isokinetic for the true airspeed reached with Polar 6. Thus, the particle transmission by the inlet can be assumed to be near unity for particles from 20 nm to about 1 μm. For larger particles, the transmission drops to to 80 % at 5 μm and 30 % at 10 μm. For the individual instruments, which have different particle size ranges and tubing to the main inlet, specific transmission efficiencies need to be considered. Outside clouds, the CVI was also used as an aerosol and gas inlet. The respective suitability was demonstrated by the excellent agreement of aerosol particle number size distributions measured behind the CVI and the Polar 6 standard aerosol inlet (Ehrlich et al., 2019). For the ambient aerosol particle and trace gas collection no enrichment exists and the overall sampling efficiency is 100 %.

#### 3.5.2 Polar 6 - Aerosol microphysical in-situ observations

A condensation particle counter (CPC, TSI-3010) and an ultra-high sensitivity aerosol spectrometer (UHSAS) were permanently connected to the CVI inlet to measure particle number concentration and size distribution (PNSD) of CPR (inside clouds) and ambient aerosol particles (outside clouds). Data of both single particle measurement devices are published by Mertes and Wetzel (2023) with a temporal resolution of 1 Hz. For the CPC, the number concentration of particles larger than 10 nm wass





measured. The UHSAS covers the number size distribution of particles with diameters between 65 nm and 1 μm. All data are provided in ambient concentration at the measurement point. The PNSD measured by the UHSAS were temporary biased by an electronic problem, causing particles to be falsely classified into two bins (107.9–111.1 nm and 211.66–217.96 nm) while the total particle number concentration is preserved. A correction scheme was developed to detect such biased PNSDs and to

redistribute the surplus of the two affected bins. A flag is included in the published data set to indicate if a spectrum is unbiased or biased and corrected.

The PNSD in the mobility diameter range of 10 nm to 800 nm were measured using a Mobility Particle Size Spectrometer (MPSS, Wiedensohler et al., 2012). During HALO–$(\mathcal{AC})^3$, the MPSS was permanently connected to the standard aerosol inlet. Near-isokinetic sampling was established for the majority of the flight time by restricting the total aerosol flow through

the sampling line (inner diameter 0.75 inch) with the help of a valve. With this, an upper cut-off diameter of about 4 μm was achieved. Each scan of the entire particle size range took approximately 5 min. Data are provided by Tatzelt et al. (2024b) as bin-wise log-normal particle number ($\mathrm{d}N/\mathrm{dlog}D_\mathrm{p}$). For each of the 40 size bins, the center particle size $D_\mathrm{p}$ is given. The start and end of each scan is included in the data set. Each PNSD was integrated over all $D_\mathrm{p}$ to obtain the total particle number concentration ($N_\mathrm{total}$). A data flag is given to indicate when sampling was under near-isokinetic conditions.

The PNSD of larger aerosol particles in the size range between 0.25 μm and 40 μm was measured by a Grimm Sky optical particle counter (Sky-OPC) model 1.129 (Heim et al., 2008; Bundke et al., 2015). The data including all 31 size bins of the log-normal PNSD and the total particle concentration are published by Eppers et al. (2023b) with a temporal resolution of 6 s. The Sky-OPC was connected either to the standard aerosol inlet or the CVI, which is indicated in the data set by a flag. The particle number concentrations are pressure-corrected to standard temperature and pressure.

### 3.5.3 Polar 6 - Cloud condensation and ice nucleating particles measurements

A miniature Cloud Condensation Nuclei (CCN) Counter (mCCNC), developed by Roberts and Nenes (2005), was used to measure CCN number concentration ($N_\mathrm{CCN}$) at a constant supersaturation (SS) of 0.1 %, according to post-campaign calibration. Data are provided by Tatzelt et al. (2024a) and include $N_\mathrm{CCN}$, SS, and activation temperature $T_\mathrm{act}$ with a temporal resolution of 1 Hz. The mCCNC was connected to either the standard aerosol inlet or the CVI. In the data set, a flag indicates which inlet

was used for each time. Data quality was ensured by taking into account additional monitoring parameters inside the mCCNC that indicate whether the instrument was within its operational window. Only for the Polar 6 research flights RF08–RF13 data quality was found to be sufficient and is published.

The High-volume flow aERosol particle filter sAmpler (HERA, Grawe et al., 2023) was deployed for offline analysis of ice nucleating particles (INP) concentrations (Chen et al., 2018; Hartmann et al., 2019). Up to 5 filters were sampled per flight at

a volumetric flow rate of 30 L$^{-1}$. One filter per flight was reserved as a blank, i.e., handled in the same way as the other filters but not exposed to the sample flow, for determination of the measurement background. Typically, filters were sampled during horizontal flight legs in different altitudes and over varying surfaces (sea ice, open ocean, land). The sampling time varied between 2 min and 140 min depending on the target area. HERA was connected to the Polar 6 standard aerosol inlet for the vast majority of the filter sampling. On rare occasions, HERA was switched to the CVI inlet to sample cloud particle residuals.





The filter samples were removed from HERA after each research flight in a clean laboratory environment and stored at -18 °C until further analysis.

### 3.5.4   Polar 6 - Light absorbing particles measurements

The properties of refractory BC (rBC) particles were measured by a single particle soot photometer (SP2) in an identical setup as used during the ACLOUD campaign (Zanatta et al., 2023). A single wavelength particle soot absorption photometer (PSAP,

Bond et al., 1999) was applied to measure the particle absorption coefficient. The refractory black carbon mass and number concentration measured by the SP2 are published by Jurányi and Herber (2024) for a sampling time of 3 s. The absorption coefficient from the PSAP is measured with a sampling time of 10 s and included in the data set published by Mertes and Wetzel (2023).

   The SP2 data were processed on basis of a calibration with standard material with known BC mass (Schwarz et al., 2006;

N. and Y., 2010). During HALO–$(\mathcal{AC})^3$, the SP2 measured the BC mass of individual BC-containing aerosol particles in the mass equivalent diameter range of 70 nm to 675 nm, assuming void-free bulk material density of 1.8 g cm$^{-3}$. The instrument was permanently connected to the CVI inlet to measure rBC properties of CPR inside clouds, and of ambient aerosol particles above, below clouds. A calibration before and after the campaign was performed using size-selected fullerene soot particles (Gysel et al., 2011; Laborde et al., 2012). Unfortunately, due to some laser problems of the SP2, no measurements exist during

the Polar 6 flight numbers RF02, RF03 and RF08. During the other flights, this issue occured only at the beginning of the flights. In the published data set a flag is provided that indicates if the CVI counterflow was switched off or on (i.e. if the instrument sampled ambient aerosols or CPR). The data are not corrected for the CVI enrichment factor.

   The PSAP was permanently connected to the CVI inlet to measure the absorption coefficient of CPR (inside clouds) and ambient aerosol particles (outside clouds), which is given in ambient conditions at the measurement point. Data inside and

outside of clouds can be distinguished by the flag provided in the CVI data set. The raw data were corrected to account for filter effects (Bond et al., 1999) except for scattering, because the filters were changed, when the transmittance was still high. The PSAP was set for a 10 s integration time, which not always sufficient, so that also negative values occurred. This means that longer averaging times might be sometimes required to get a meaningful value for the absorption coefficient. By use of a mass absorption efficiency, the absorption coefficient can be converted in to a black carbon (BC) mass concentration, whereby

one should clarify that mineral dust did not significantly contribute to the particle absorption. For the PSAP a mass absorption efficiency of 14.7 m$^2$g$^{-1}$ was derived in a previous study (Mertes et al., 2004).

### 3.5.5   Polar 6 - Aerosol chemistry in-situ observations

The chemical composition of the aerosol particle and cloud particle residual populations was measured using the single particle laser ablation mass spectrometer ALABAMA (Aircraft-based Laser ABlation Aerosol MAss spectrometer, Brands et al.,

2011). Compared to previous campaigns (Ehrlich et al., 2019), the detectable particle size range of the ALABAMA was extended to larger particle diameters by modifications reported by Clemen et al. (2020) and ranges now between 230 nm and 3000 nm. The processed data are published by Eppers et al. (2023a) and specifies the size and chemical composition of





individual particles. To measure both aerosol particle and cloud particle residuals, the ALABAMA was mostly operated behind the CVI with the counterflow switched on inside clouds and off outside clouds. For specific flight segments, the ALABAMA

was also operated alternately behind the standard aerosol inlet and the CVI to investigating potential differences between the two inlets. Details on this procedure are explained by Ehrlich et al. (2019). The sampling conditions are indicated in the data set by a flag. As an indicator for measurements inside clouds, the inlet position of the switching valve at the ALABAMA, as well as the supply flow data from the CVI were used.

During HALO–$(\mathcal{AC})^3$, a total of about 137000 mass spectra were detected (132000 aerosol particles and 5000 cloud residu-

als). The data processing/evaluation of the single particle mass spectra recorded with the ALABAMA was carried out using the CRISP software package (Concise Retrieval of Information from Single Particles, Klimach, 2012). To quantify the chemical composition of the particle population, the shape of the mass spectra was analyzed and assigned to a cluster (particle type). A fuzzy-c-means algorithm was used for grouping mass spectra with similar signal patterns (Roth et al., 2016). In contrast to Roth et al. (2016), the threshold correlation coefficient above which a mass spectrum was assigned to one of the clusters was

reduced to 0.6. Below this threshold, the particle was classified in the "others" cluster. This reduced the need for subsequent manual inspection of the residual cluster. In addition, a "startcluster difference" of 0.9, a "fuzzifier" of 1.4, and a "fuzzy abort" of $1\mathrm{e}^{-4}$ were used (cf. Roth et al., 2016). The classification initially considers 50 clusters. This was reduced to 37 clusters (particle types) based on manual inspection of similarities in the mass spectrum and in their temporal occurrence. The published data set provides for each particle the detection time, assigned particle types/cluster (37 specific types and "others"), and

the vacuum-aerodynamic particle diameter.

### 3.6   Polar 6 - Trace Gas Chemistry

Trace gases carbon monoxide (CO), carbon dioxide ($CO_2$), water vapor ($H_2O$) and Ozone ($O_3$) were measured using an ultrafast CO monitor model AL5002 for CO (Gerbig et al., 1999), a LI-7200 closed $CO_2$/$H_2O$ analyser for $CO_2$ and $H_2O$ (Burba et al., 2010) and a Dual Beam Ozone Monitor model 2BTech-205 for $O_3$ (Johnson et al., 2014). All data are published by

Bozem et al. (2024) with 1 Hz resolution, except for $O_3$, which was measured at 0.5 Hz resolution.

The instruments were connected to a specific backward facing trace gas inlet using teflon tubing. Venturi Valves mounted on the fuselage in the back of the aircraft maintained a flow of 10-20 slm within the inlet lines and the instruments sampled from the main inlet lines using a T-type insertion. Using NIST traceable standards with CO and $CO_2$ concentrations at atmospheric levels and $H_2O$ concentrations close to zero, every 15-30 min, regular in-situ calibrations were performed during the flights to

correct for any instrument drift. Details on this calibration procedures are described by Bozem et al. (2019).

In parallel, a water vapor sensor system (WVSS-II) based on a tuneable diode laser (TDL) was operated onboard of Polar 6. The system was permanently connected to the CVI inlet to measure the cloud water content (CWC) (inside clouds) and the ambient water vapor mixing ratio (outside clouds). Data are published by Mertes and Wetzel (2023) and include CWC and $H_2O$ mixing ratio defined as ambient concentrations at the measurement point. The WVSS-II has a lower detection limit of

50 ppmv and was operated at a 1 Hz temporal resolution. During the in-cloud sampling of cloud particles, the liquid and frozen phase from all collected hydrometeors were driven into the gas phase by the CVI inlet (Ogren et al., 1985). Thus, the former



condensed water was measured as water vapor behind the CVI, which was then converted into the sampled CWC (Mertes et al., 2001).

## 4 Data availability and data access

**Table 4.** Available data sets from measurements on HALO during HALO–$(\mathcal{AC})^3$ collected by Ehrlich et al. (2024a).

| HALO Instrument | PANGAEA dataset ID | Format | Reference |
| --- | --- | --- | --- |
| BAHAMAS | HALO Data Base | netcdf | Giez et al. (2022) |
| Master tracks | 967299 | ascii | Ehrlich et al. (2024b) |
| Lagrangian Matches 1-day | 967143 | ascii | Kirbus et al. (2024a) |
| Lagrangian Matches 2-days | 967148 | ascii | Kirbus et al. (2024b) |
| SMART | 956151 | netcdf | Röttenbacher et al. (2023) |
| VELOX | 963401 | netcdf | Schäfer et al. (2023) |
| KT19 | 967378 | netcdf | Schäfer et al. (2024) |
| BACARDI | 963739 | netcdf | Luebke et al. (2023) |
| HAMP Unified | 963250 | netcdf | Dorff et al. (2023) |
| WALES | 967086 | netcdf | Wirth and Groß (2024) |
| specMACS SWIR | 966992 | netcdf | Weber et al. (2024b) |
| specMACS polarization cameras | 965546 | mp4 | Weber et al. (2024a) |

All datasets are published in PANGAEA with open access. Only data from BAHAMAS on HALO are published at the HALO database repository for consistency reasons. Tables 4, 5, and 6 list the corresponding dataset identifiers. Within PANGAEA, aircraft-separated dataset collections of all corresponding datasets were compiled for HALO (Ehrlich et al., 2024a, https://doi.org/10.1594/PANGAEA.968885), Polar 5 (Mech et al., 2024a, https://doi.org/10.1594/PANGAEA.968883), and Polar 6 (Herber et al., 2024, https://doi.org/10.1594/PANGAEA.968884). To provide access to the objective of each research

flight, the motivation for the specific flight pattern, and a brief overview of the current weather conditions, flight reports were compiled and published by Ehrlich et al. (2024). These reports also provide notes of the instrument operators, in-flight images, and quick-looks, when available.

With the exception of some instruments available in compressed ASCII format, all datasets have been converted to and are available in NetCDF4 file format. In general, each data file contains the data for one research flight. The files are identified

by aircraft identifier ("HALO", "P5", "P6"), instrument short name, date, and research flight number according to Table 1. An exception is the data of the fish-eye camera. With several gigabytes per hour, these data are very large and therefore provided in hourly files.





**Table 5.** Available data sets from measurements on Polar 5 during HALO–$(\mathcal{AC})^3$ collected by Mech et al. (2024a).

| Polar 5 | | | |
| --- | --- | --- | --- |
| Instrument | PANGAEA dataset ID | Format | Reference |
| Master tracks | 947788 | ascii | Herber et al. (2022a) |
| Noseboom (1 Hz) | 968911 | ascii | Lüpkes et al. (2024b) |
| Noseboom (100 Hz) | 968952 | ascii | Lüpkes et al. (2024a) |
| SMART | 963112 | netcdf | Jäkel et al. (2023a) |
| Broadband Radiometer | 963654 | netcdf | Becker et al. (2023) |
| Fish-Eye camera radiance | 967288 | netcdf | Jäkel and Wendisch (2024) |
| Fish-Eye camera surface product | 962996 | netcdf | Jäkel et al. (2023b) |
| AMALi cloud mask and cloud top altitude | 964985 | netcdf | Mech et al. (2024d) |
| MiRAC-A radar reflectivities and brightness temperatures | 964977 | netcdf | Mech et al. (2024b) |
| HATPRO brightness temperatures | 964982 | netcdf | Mech et al. (2024c) |
| Aisa EAGLE | 967347 | netcdf | Klingebiel et al. (2024) |

**Table 6.** Available data sets from measurements on Polar 6 during HALO–$(\mathcal{AC})^3$ collected by Herber et al. (2024).

| Polar 6 | | | |
| --- | --- | --- | --- |
| Instrument | PANGAEA dataset ID | Format | Reference |
| Master tracks | 947701 | ascii | Herber et al. (2022b) |
| Noseboom (1 Hz) | 968911 | ascii | Lüpkes et al. (2024b) |
| Noseboom (100 Hz) | 968952 | ascii | Lüpkes et al. (2024a) |
| Broadband Radiometer | 963654 | netcdf | Becker et al. (2023) |
| CDP, CIP, PIP | 963247 | netcdf | Moser et al. (2023) |
| 2D-S, PN | 965734 | netcdf | Dupuy et al. (2024) |
| BCDP | 963614 | netcdf | Lucke et al. (2023) |
| Nevzorov | 963628 | netcdf | Lucke et al. (2024) |
| UHSAS, CPC, PSAP, WVSS-II | 963771 | ascii | Mertes and Wetzel (2023) |
| miniCCNC | 969122 | ascii | Tatzelt et al. (2024a) |
| MPSS | 968633 | ascii | Tatzelt et al. (2024b) |
| Trace gases | 968545 | ascii | Bozem et al. (2024) |
| SP2 | 963718 | ascii | Jurányi and Herber (2024) |
| ALABAMA | 963290 | netcdf | Eppers et al. (2023a) |
| Sky-OPC | 963284 | netcdf | Eppers et al. (2023b) |



The datasets available on PANGAEA contain all the information needed to work with the data. If not included within the respective dataset for the instruments, the aircraft position and altitude are provided in the aircraft master tracks Herber et al. (2022a, b); Ehrlich et al. (2024b).

To facilitate the use of the data, the *ac3airborne* (Mech et al., 2022c) Python module and a collection of codes are provided within in an online Jupyterbook *How to ac3airborne* (Mech et al., 2022b). Within *How to ac3airborne*, the usage of the data sets is described with Python code snippets. It also includes an overview of the specific data availability of the individual instruments. Most data sets will be automatically downloaded from PANGAEA based on entries in an intake catalog. A central part of the *ac3airborne* package is the flight segmentation. For each research flight, start and end time stamp of specific sections are listed to easily extract the data of interest. These flight sections differentiate between take-off, landing, ascents, and descents, specific patterns for in-situ probing, high, mid, or low-level legs, and patterns for calibration purposes. The usage of the segmentation is described in *How to ac3airborne*.

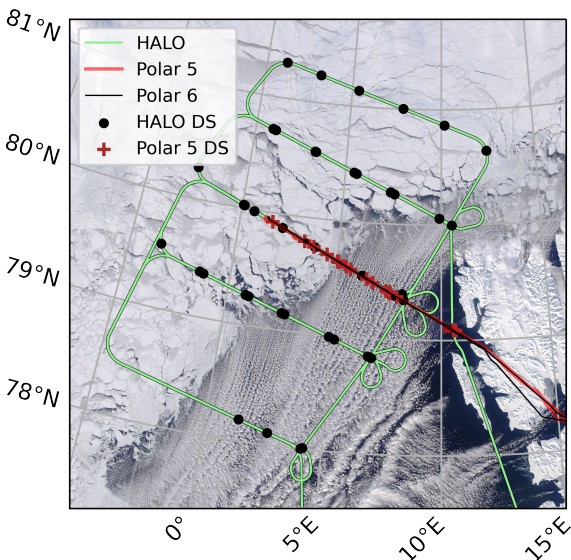

**Figure 4.** Satellite image (MODIS) and flight tracks of HALO, Polar 5 and Polar 6 for research flight on 1 April 2022. Dots show the location of dropsonde (DS) releases from HALO, crosses the dropsondes released from Polar 5. Close collocation of all three aircraft was realized along the standard leg.

## 5   Potential of combining HALO–$(\mathcal{AC})^3$ data sets

The aircraft measurements of HALO–$(\mathcal{AC})^3$ were partly sampled with similar and/or complementary instrumentation and in different altitudes. The following examples illustrate the added value of combining individual data sets to enhance the potential of data analysis. All data presented here were measured on 1 April 2022 when all three aircraft aimed to characterize


a CAO west of Svalbard. A satellite image and the flight tracks are shown in Fig. 4. While almost cloud-free conditions were present over sea ice, roll clouds formed immediately over the open ocean after the cold air mass crossed the sea ice edge.

The flight pattern of all three aircraft aimed at characterizing this air mass transformation, particularly the development of the thermodynamic profiles and cloud properties (Kirbus et al., 2024). While HALO mapped the larger area, Polar 5 and Polar 6 focused on the transition between open ocean and sea ice along the standard leg (Schirmacher et al., 2024). The standard leg was flown six times by Polar 5 and Polar 6 operating in close collocation. For three of these legs, HALO overflights were synchronized with Polar 5. The exact overpass was timed approximately when Polar 5 was in the center of the leg. This

coordination of the aircraft allowed to combine data sets from the different platforms.

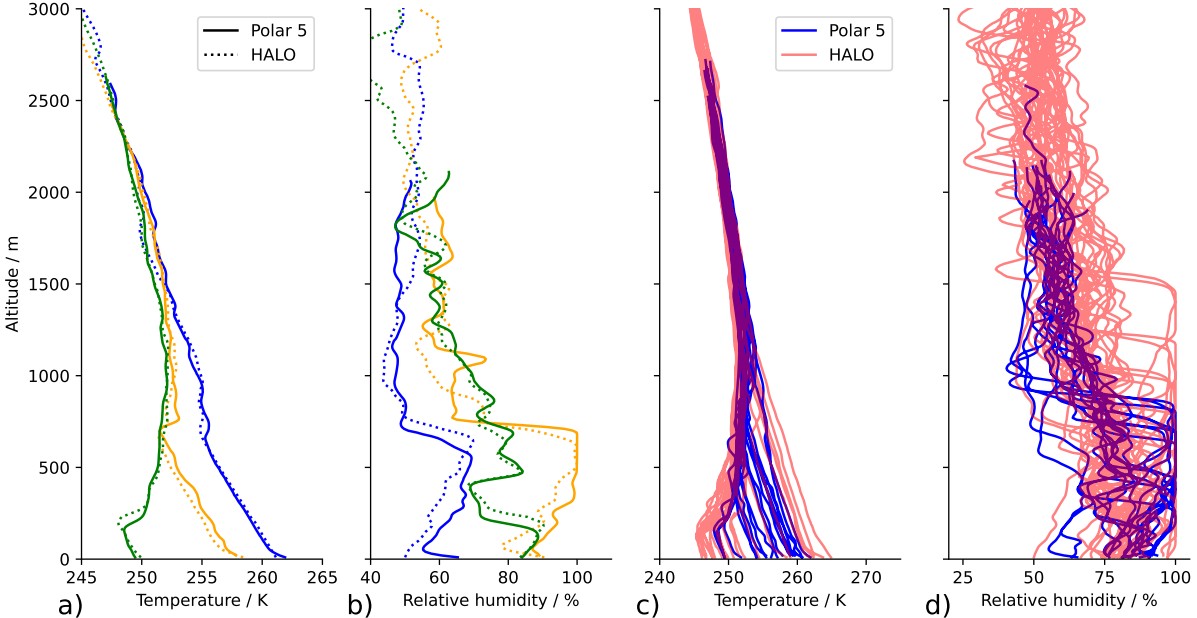

**Figure 5.** Air temperature and relative humidity profiles measured by dropsondes from HALO and Polar 5 on 1 April 2022. Panels (a) and (b) compare three pairs of closely collocated profiles from HALO (dotted) and Polar 5 sondes (solid). Panels (c) and (d) display the merged full data sets from both aircraft.

## 5.1 Merged dropsonde data

On 1 April 2022, 41 dropsondes were released from HALO and 18 from Polar 5. The locations of the dropsondes covering the transition from sea ice to open ocean are shown in Fig. 4. Due to its extended flight range, the HALO dropsondes are distributed over a larger area, while Polar 5 enhanced the spatial resolution of dropsonde profiles along the standard leg, especially in the

MIZ.

To merge the dropsonde data from both aircraft into one dataset, it is essential that the quality of both datasets is comparable. This was ensured through the data processing and quality controls explained in Section 3.1.3. In Figure 5a the agreement of



three HALO dropsondes with the closest Polar 5 dropsondes is analyzed. For each pair of dropsondes, their spatial distance is less than 6 km (at ground), the time between the releases is between 15 min and 45 min. While HALO dropsondes typically sampled the entire troposphere, Polar 5 dropsondes were limited to altitudes below 3 km. However, this is sufficient enough to study transformation processes in the atmospheric boundary layer. The temperature profiles agree within the general uncertainty of the sensors, while the humidity profiles differ slightly, due to the high spatial and temporal variability of the humidity in cloudy conditions.

Assuming steady conditions of the CAO, the air mass transformation on 1 April 2022 is analyzed with an Eulerian approach merging all dropsonde observations. Figure 5c and 5d present all available soundings obtained during this flight. The spread of the temperature profiles illustrates how the stable surface inversion over sea ice transforms into a well-mixed boundary layer over the open ocean. This transformation occurs over short distances of some 100 kilometers and is under-sampled by the HALO dropsondes. The majority of HALO dropsondes were released in larger distances to the sea ice edge either over closed sea ice or open ocean. This results in a bimodal distribution of temperature profiles in the HALO measurements. Adding the Polar 5 observations fills this gap and significantly improves the spatial resolution in the MIZ.

Although the dropsondes were released at different times, Figure 5 assumes temporally stable conditions. However, comparing dropsonde profiles, which are linked in time by air mass trajectories (cf. quasi-Lagrangian matches discussed in Section 3.1.5) allows to derive vertically resolved air temperature and humidity tendencies and to quantify the air mass transformation (Kirbus et al., 2024). Merging the HALO and Polar 5 dropsonde data sets significantly increases the number of quasi-Lagrangian matches.

## 5.2 Combining cloud radar observations from different platforms

Coordinated flights between HALO, Polar 5, and Polar 6 offer the unique opportunity to combine collocated multiple frequency cloud radar and in-situ measurements. The data collected during HALO–$(\mathcal{AC})^3$ can therefore be used to gain a better understanding of the vertical structure of Arctic clouds and investigate microphysical cloud processes. For the 1 April 2022 CAO case, Figure 6 illustrates the potential of combining effective radar reflectivity $Z_e$ observations obtained with the 35 GHz cloud radar on HALO (Fig. 6a) and the 94 GHz cloud radar MiRAC on Polar 5 (Fig. 6b). The data are displayed on the same spatial grid. Due to the faster flight speed of HALO, Figure 6a covers a shorter time range than Figure 6b. The exact overpass occurred at 10:22 UTC when both aircraft flew over the cloud-free sea ice (not shown in Fig. 6). Both radars observed similar cloud structures with cloud top altitude increasing with distance to the sea ice edge. High $Z_e$ indicate updraft regions with increased formation and growth of ice particles. These updraft regions of the roll clouds are located at almost the same positions.

A pixel by pixel comparison is only possible for these HALO-AC3 cases where a very close collocation of both aircraft was achieved. For the 1 April 2022 case instead, a comparison by means of averaged $Z_e$ vertical profiles is done as an alternative and shown in Fig. 6c. Near the cloud top, the $Z_e$ measured at 35 GHz on HALO and 94 GHz on Polar 5 overlap indicating that the cloud particles are in the Rayleigh scattering regime at both frequencies, corresponding to small particles less than 0.3 mm in diameter. In lower cloud layers, the 35 GHz radar measured higher $Z_e$ than the 94 GHz radar indicating larger, denser particles that are in the Mie scattering regime at 94 GHz but likely still in the Rayleigh regime at 35 GHz (i.e., particles
smaller than 0.9 mm). The increase in particle size and density in lower cloud layers indicates particle growth processes such as riming, where liquid droplets freeze onto ice particles. Based on collocated Polar 5 and Polar 6 flight segments during HALO–$(\mathcal{AC})^3$, Maherndl et al. (2024a, b) found riming to occur frequently for over 85 % of the measured ice particles so that riming

contributed on average 63 % to the IWC. During the flight segment of the 1 April case analyzed here, Polar 6 was flying close to the ground collecting in-situ measurements of the precipitation below cloud. Manual inspection of in-situ images indeed reveals indications of riming (not shown). This example demonstrates, that in future studies, dual-wavelength ratios computed for coordinated flight segments have the potential to study ice formation and growth processes in dependence of the cloud vertical structure.

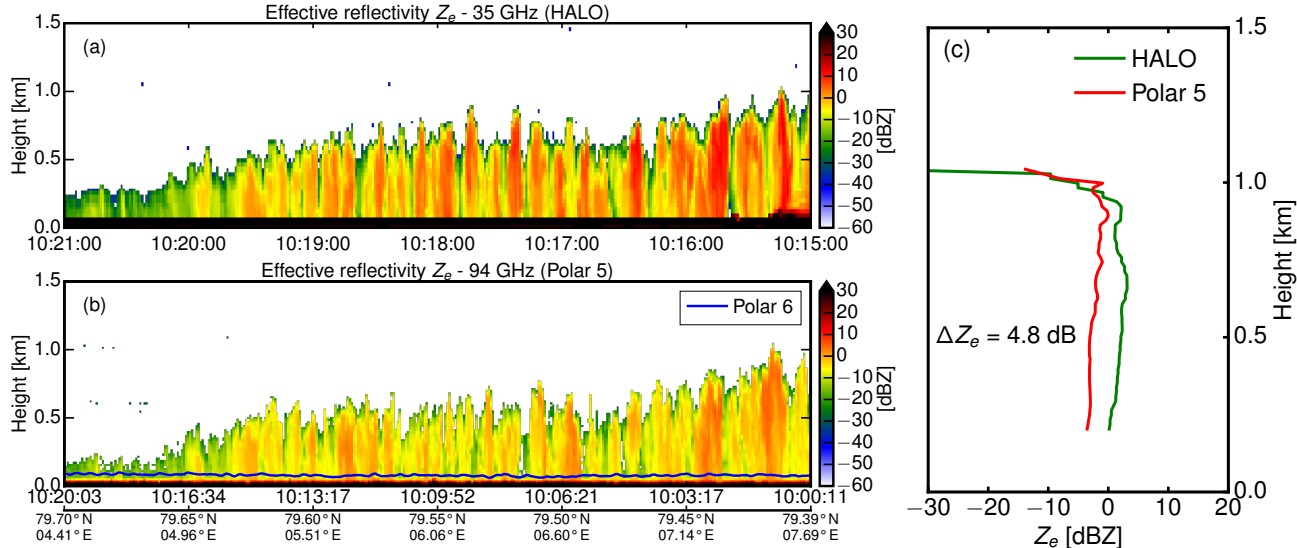

**Figure 6.** Collocated flight between HALO, Polar 5, and Polar 6 on 1 April 2022 to combine observations from (a) the 35 GHz cloud radar on HALO with (b) the 94 GHz cloud radar MIRAC on Polar 5. The mean profiles of dual-wavelength ratio between both radars in (c) are used to interpret in-situ measurements from Polar 6 of snow microphysical properties.

### 5.3 Passive remote sensing in different scales and spectral ranges

The passive spectral imagers on board of HALO (specMACS, VELOX) and Polar 5 (AISA Eagle/Hawk) provide observations across different spectral ranges and resolve clouds and surfaces with different spatial resolutions. Merging these data sets enables the combination of spectral information and the investigation of cloud inhomogeneities on different scales. An example of observations in the MIZ is shown in Fig. 7 for the CAO case observed on 1 April 2022. For two flight sections, one above

685 sea ice (I, with 80 % sea ice cover) and one over open ocean (II, without sea ice), data measured by all three imagers are combined. The images are collocated by georeferencing each pixel using the geometric calibrations of the optics and aircraft attitude information. For specMACS and AISA Hawk, the image shows spectral radiances at a similar wavelength of about 1200 nm. While specMACS is able to capture surface and cloud structures (bright updraft and darker downdraft regions) with



Data

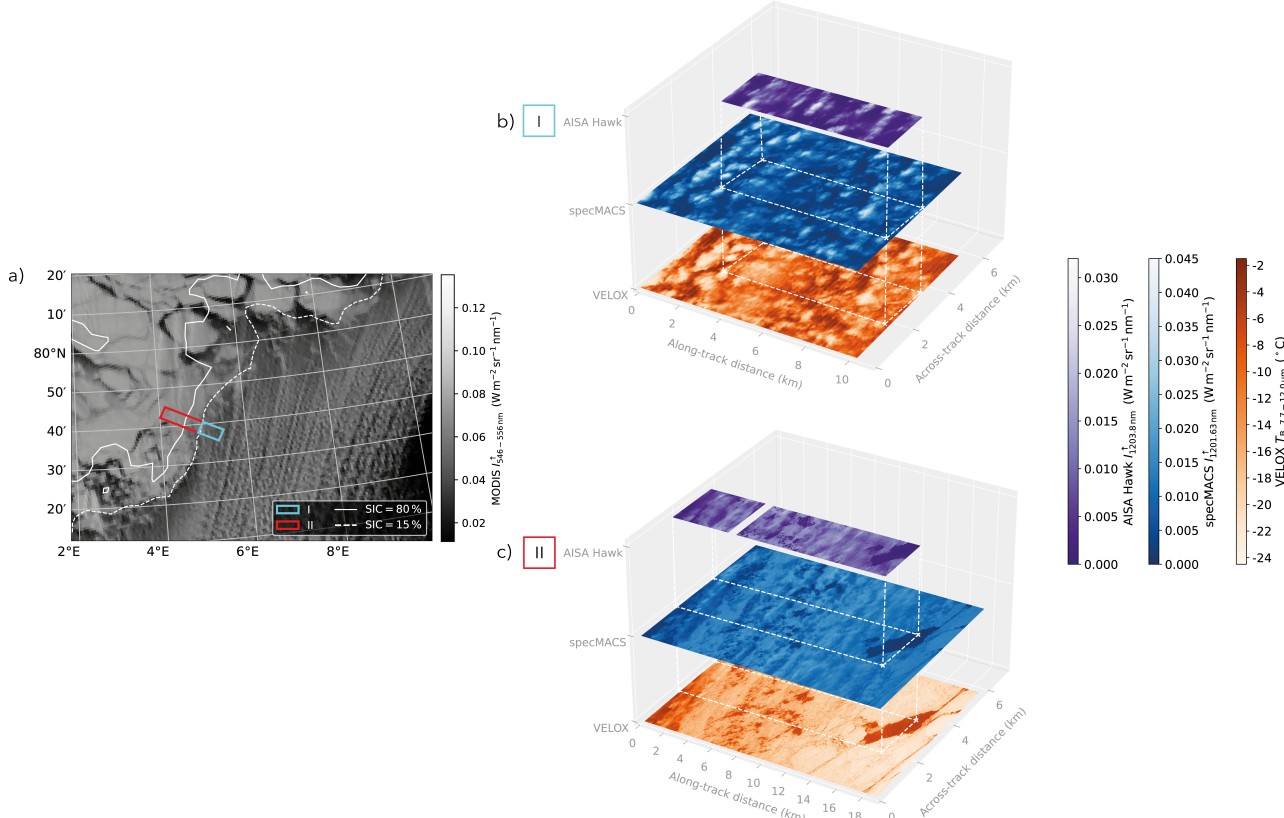

**Figure 7.** Combined data examples from the solar and thermal-infrared imagers onboard of HALO (specMACS and VELOX) and Polar 5 (AISA Hawk). (a) MODIS satellite image indicating the area of the two example segments I and II for which spectral radiance and brightness temperatures are shown in panel (b) (segment I) and (c) (segment II). For specMACS and AISA Hawk radiance at 1.2 μm are shown and merged with the brightness temperature of the broadband channel of VELOX (7.7-12 μm).

18 m pixel size (across-track), the lower flight altitude of Polar 5 allows AISA Hawk to resolve finer scales 5 m pixel size.
Even single ice floes can be identified in the cloud-free areas. With help of these different resolutions, 3D radiative effects and the scale dependence of surface and cloud retrieval can be investigated.

The two imagers on HALO have almost the same spatial resolution and field of view. Thus, VELOX and specMACS observations can combine radiances in the solar and thermal-infrared spectral range. Figure 7 shows the broadband thermal-infrared channel of VELOX (7.7-12 μm) and the shortwave infrared channel of specMACS (1.2 μm). Over open water, VELOX
brightness temperatures are particularly sensitive to the emission of thin clouds, e.g., the filaments in the downdraft region that often are identified as cloud-free in specMACS. Over sea ice, where the clouds are in their developing stage and thus very low, surface and cloud top temperature are often similar and cannot be distinguished in the VELOX measurements. However, the different spectral absorption of the liquid clouds compared to sea ice makes the clouds detectable in the specMACS image. By



combining different spectral ranges, the observations can mimic the most common satellite imaging spectrometers. Satellite

retrieval algorithms can then be applied to the airborne data to validate satellite products of surface and cloud properties.

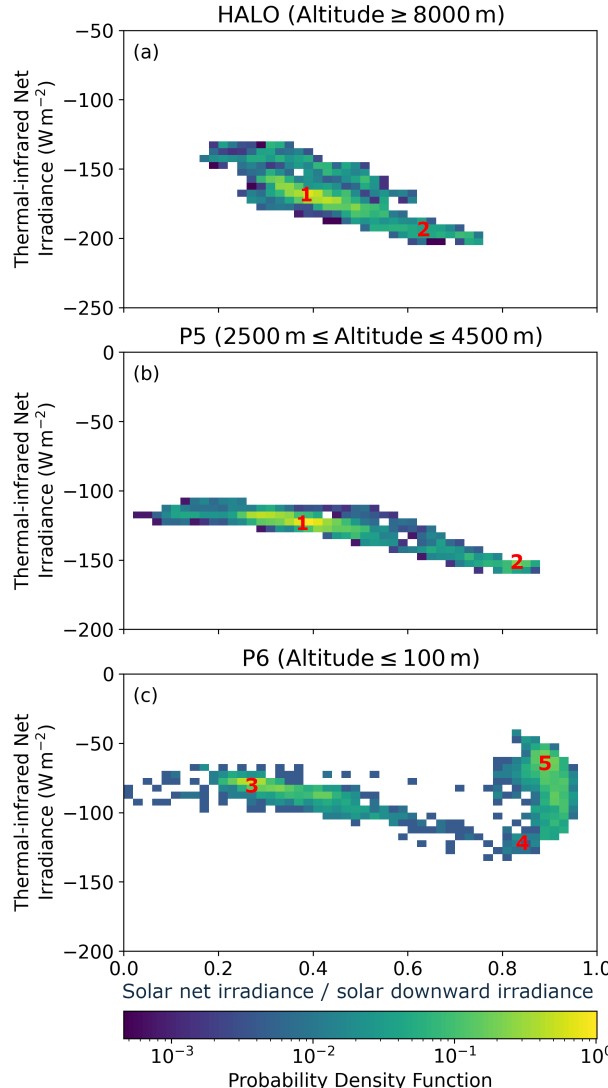

**Figure 8.** Two-dimensional probability density function of the solar (normalized by the solar downward irradiance) and thermal-infrared net irradiances observed on 1 April 2022 in different altitude regimes: (a) HALO, (b) Polar 5, and (c) Polar 6 flight altitudes. Mode numbers: 1 – above clouds or sea ice, 2 – above open ocean, 3 – cloud-free over sea ice, 4 – cloud-free over open ocean, 5 – cloudy over open ocean.





## 5.4 Altitude dependence of the radiant energy budget

The radiant energy budget (REB) in the Arctic depends on various surface, cloud, and thermodynamic properties (Sedlar et al., 2011; Stapf et al., 2021). At the surface, Wendisch et al. (2023b) identified four modes (regimes) of the REB, determined by the surface type (sea ice or open ocean) and the atmospheric state (cloudy or cloud-free conditions). During HALO–$(\mathcal{AC})^3$, the

REB, quantified by the net irradiance, was measured on board of all three aircraft with a similar set of broadband radiometers. The coordinated combination of the three aircraft provides measurements in different altitude regimes. While Polar 6 sampled the REB close to the surface, the REB above the Arctic boundary layer clouds and close to the tropopause was observed by Polar 5 and HALO, respectively. Combining these data sets allows for an extended statistical analysis of the Arctic REB to higher altitudes.

For the 1 April 2022 CAO case, Figure 8 illustrates the altitude dependence of the thermal-infrared REB (y-axis) and the solar REB normalized by its downward component (x-axis). The common mode structure (cloudy and cloud-free, over sea ice and open ocean) is most prominently illustrated in the near-surface REB (modes indicated by 3, 4, and 5, Fig. 8c). Modes 4 and 5 result from the observations in cloud-free and cloudy conditions above Polar 6 when flying over the open ocean. Mode 3 corresponds to cloud-free situations over sea ice (only few clouds were observed over sea ice). For Polar 5 (mid-level regime,

Fig. 8b), clouds were not present above the aircraft, leaving a combined mode for weaker emitting cases above sea ice or low clouds (mode 1) and a mode for cloud-free observations above the darker open ocean (mode 2). The REB in high altitudes (Fig. 8a) shows a continuous transition between modes 1 and 2 and less extreme values due to the averaging of the larger field of view. In general, the thermal-infrared net irradiance becomes more negative with increasing altitude. This is mostly caused by the reduction of the downward irradiance emitted by the atmosphere above the aircraft, as the effective brightness temperature

decreases for higher altitudes. Numerical models often struggle to correctly represent this mode structure due to limitations in the treatment of sub-grid processes including clouds and the sea ice albedo (Kretzschmar et al., 2020; Solomon et al., 2023). Consequently, the combined HALO–$(\mathcal{AC})^3$ measurements are valuable for identifying potential misrepresentations of properties affecting the REB.

## 6 Conclusions

The HALO–$(\mathcal{AC})^3$ campaign provides a comprehensive in-situ and remote sensing observational data set characterizing Arctic air mass transformations during warm air intrusions (WAIs) with on-ice flow and cold air outbreaks (COAs) with off-ice flow. The data set comprises measurements from three research aircraft, HALO, Polar 5, and Polar 6, each operating at different altitudes and with different spatial coverage. All data are published in the PANGAEA database by instrument-separated data subsets. This paper provides an overview of these data sets, the campaign specific instrument operation, data processing, and

data quality. For detailed information, respective references are provided. To facilitate the quasi-Lagrangian analysis of the airborne data, the locations of quasi-Lagrangian matches are published. As it is important for potential model evaluations, an overview of the data transferred to the Global Telecommunication System (GTS) and assimilated by the ECMWF services is given.





It is highlighted how the scientific analysis of the HALO–$(\mathcal{AC})^3$ data benefits from the coordinated operation of three
aircraft. For a CAO case, it is shown, that the higher spatial resolution of measurements by Polar 5 and Polar 6 in a smaller
area close to the marginal sea ice zone (MIZ) complements the less resolved but wider spread observations of HALO. Merging
the dropsonde data provides an almost continuous view on the evolution of cold air masses during a CAO. The radiative
energy budget and clouds and surface properties can be resolved in different scales by remote sensing instruments on HALO
and Polar 5. Synergistic effects of multi-frequency radar remote sensing and multi-wavelength passive remote sensing are
demonstrated, e.g. riming of ice particles is identified within the clouds of the CAO.

A series of ongoing studies have already made use of the HALO–$(\mathcal{AC})^3$ data, concentrating on some of the highlights
presented by Wendisch et al. (2024). These studies are collected in the inter-journal special issue of Atmospheric Chemistry
and Physics and Atmospheric Measurement Techniques, "HALO–$(\mathcal{AC})^3$ – an airborne campaign to study air mass transfor-
mations during warm-air intrusions and cold-air outbreaks" (https://acp.copernicus.org/articles/special_issue1272.html, last ac-
cess: 11 March 2024). However, the data set has a lot of further potential for detailed studies of the evolution of thermodynamic
and cloud properties in air mass transformations, the Arctic moisture and radiative energy budget, cloud–aerosol interaction,
improvement of satellite remote sensing, mesoscale dynamics in the Arctic, validation of cloud resolving numerical models,
and more. The use of HALO–$(\mathcal{AC})^3$ data is encouraged as a continuation of a series of previous aircraft campaigns with almost
identical instrumental setup. Data from ACLOUD, AFLUX, and MOSAiC-ACA (all data published on PANGAEA, Ehrlich
et al., 2019; Mech et al., 2022a) complement the HALO–$(\mathcal{AC})^3$ observations for different seasons and sea ice conditions and
allow for statistically solid analyses of atmosphere, cloud, aerosol, trace gas, and sea ice properties. Further data products that
are currently in development will be added to PANGAEA in future and will be linked to the current data set within PANGAEA
via the tag "HALO–$(\mathcal{AC})^3$".

## 7 Code and data availability

Each instrument is controlled either by code developed by the institution operating it or by the manufacturer and, therefore,
often closed source or not even freely available and bundled with the instrument. Below, the public available software are listed.

The *ac3airborne* package and tools developed within the project are written in Python, open source, and publicly available
on GitHub (Mech et al., 2022c). On Polar 5, MiRAC-A radar data have been corrected for nadir view by Python-based package
available on github (Mech et al., 2024). On Polar 6, the SP2 data were analyzed using the wavemetrics IGOR pro 7 toolkit (SP2
toolkit 4.115) available by Gysel-Beer and Corbin (2019). On HALO, HAMP brightness temperature offsets were corrected
with the post-correction code, which is available on GitHub (Walbröl, 2024). The dropsondes' raw data from Polar 5 and
HALO are post-processed with the Atmospheric Sounding Processing ENvironment(ASPEN Version 4.0.0, Martin and Suhr,
2024).

Data availability is summarized in Tables 4, 5, and 6 of the main text.



*Author contributions.* M.We., S.C., M.Me., A.H., M.K., C.L., S.Bo., and A.E. conceived the flight missions. A.E. designed the structure of the manuscript and merged and harmonized the contributions of the individual instrument groups. M.B. maintained and provided the infrastructure for quick-looks and data exchange during the campaign and is responsible for the data management of the $(\mathcal{AC})^3$ project. The instrument operation, data processing, and publication as well as the description provided in the corresponding sections were organized by the individual groups as follows:

On HALO, A.G. was responsible for the basic data acquisition system BAHAMAS. M.S. and S.R., were responsible for VELOX and KT-19. A.L. and J.R. were responsible for BACARDI and SMART. H.D. was responsible for HAMP - Passive and F.E. for the calibration of HAMP - Active. H.D. prepared the HAMP unified data set, with contributions from A.Wa. and F.E.. V.P. and A.We. were responsible for specMACS. M.Wi. was responsible for the WALES lidar. On Polar 5, C.L. and J.H. were responsible for the basic meteorological data and nose boom measurements. S.Be., E.J., H.M., and M.K. were responsible for broadband radiometer, AISA-Eagle/Hawk, SMART, Nikon and

KT-19. M.Me., I.S., Sa.S. and N.R. were responsible for MiRAC-A, HATPRO, and AMALi. On Polar 6, C.L. and J.H. were responsible for the basic meteorological data and nose boom measurements. R.D., C.G., O.J., and G.M. were responsible for the 2D-S and PN instruments. J.M., E.D.L.T.C., C.V., J.L., and M.Mo. were responsible for the BCPD, Nevzorov probe, CDP, CIP, and PIP. S.M. and B.W. were in charge for CVI inlet and the CPC, UHSAS, PSAP, and WVSS-II instruments. Z.J. and A.H. were responsible for the SP2 measurements. F.S., J.Scha. and S.G., were responsible for HERA and the off-line INP measurements. G.R. and C.T. were responsible for the mCCNC. H.-C.C.,

O.E., J.Schn., and P.J. were responsible for the ALABAMA and the Sky-OPC. H.B. was responsible for the trace gas measurements and got support by H.-C.C., O.E., J.Schn., and P.J..

The dropsonde launches on Polar 5 and HALO were coordinated by G.G, M.Me, and A.L.. G.G. was responsible for post-processing and publishing the data set with support of M.K.. A.S. evaluated the assimilation of dropsonde data in GTS and wrote Section 3.1.4. B.K. performed the quasi-Lagrangian analysis by trajectory calculations and contributed to Section 3.1.5. The combined analysis of dropsonde

measurements in Section 5.1 was contributed by M.K, G.G., and A.E. Section 5.2, the analysis to combine cloud radar observations from different platforms, was compiled by F.E., M.Ma., M.Me., and N.M. The comparison of the imaging spectrometers in Section 5.3 was lead by S.R., M.S. and A.E. All authors (except H.B., M.B., C.G., O.J., G.R., S.R., A.S., and C.T.) conducted the field campaign. All authors revised the manuscript.

*Competing interests.* The authors declare that they have no conflict of interest.

*Acknowledgements.* We gratefully acknowledge the funding by the Deutsche Forschungsgemeinschaft (DFG, German Research Foundation) – project number 268020496 – TRR 172, within the Transregional Collaborative Research Center "ArctiC Amplification: Climate Relevant Atmospheric and SurfaCe Processes, and Feedback Mechanisms (AC)³". The authors are grateful to AWI for providing and operating the two Polar 5 and Polar 6 aircraft. We thank the crews and the technicians of the three research aircraft for excellent technical and logistical support. The generous funding of the flight hours for the Polar 5 and Polar 6 aircraft by AWI is greatly appreciated. We are further grateful for

funding of project grant number 316646266 by DFG within the framework of Priority Program SPP 1294 to promote research with HALO. J.L. received funding from the European Union's Horizon 2020 research programme under grant agreement N° 82425 (SENS4ICE), J.M. and M.Mo. received funding from the Deutsche Forschungsgemeinschaft (DFG) – TRR 301 – Project-ID 428312742 and E.D.T.C. and C.V. from DFG SPP 1294 HALO under contracts no VO 1504/7-1 and VO 1504/10-1 no 522359172.





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
