# Peer review of "A comprehensive in-situ and remote sensing data set collected during the HALO- $(\mathcal{AC})^3$ aircraft campaign"

_Earth System Science Data, 2024_

## Author Comment (AC1)

**Reply to RC1: 'Comment on essd-2024-281', Anonymous Referee #1, 30 Jul 2024**

Thanks for the detailed and constructive review. The comments of the reviewer have been helpful to improve the manuscript. We are especially thankful for pointing at the missing links to the accompanying studies and data sets of other HALO-$(AC)^3$ activities, which significantly increased the value of the manuscript for potential readers.

Our detailed replies on the reviewers' comments are elaborated below. The reviewers' comments are given in bold, while our replies are in regular roman letters. Citations from the revised manuscript are given in italic fonts.

**Some general remarks:**

**A brief overview of the prevailing meteorological conditions in the Atlantic Arctic during spring 2022 would enhance the paper's context. Was that year exceptional or "on average" with respect to temperatures, occurrence of CAOs and WAIs? (I didn't find such a description in Wendisch et al., 2024 or elsewhere. If I overlooked, please provide a corresponding reference.) It could be easily introduced towards the end of chapter 1 or as a dedicated paragraph in chapter 2.**

> Thanks for identifying this gap in our paper. Indeed, there is a third overview paper for the HALO-$(AC)^3$ campaign (Walbröl et al., 2024) that puts the campaign into a climatological context. Walbröl et al. (2024) also describes the general synoptic conditions during the research flights. Reanalysis data and continuous ground-based observations were analyzed for this purpose.
>
> In the revised manuscript we explicitly referred to this synoptic overview paper. We have implemented the following text:
>
> > *"These synoptic events, the general meteorological conditions, and the sea ice distribution during HALO-$(AC)^3$ are summarized and compared to the long-term climatology by Walbröl et al. (2024). They found that two WAI events were associated with an extraordinary strong moisture transport that led to record- breaking near-surface air temperatures and precipitation rates in Svalbard. Although these WAI events were followed by one of the longest CAO event on record, the entire campaign period was warmer than the climatological mean."*
>
> > *"To identify how the flights were affected by the general meteorological and sea ice conditions during HALO-(AC)³, the reader is referred to Walbröl et al. (2024)."*

Walbröl, A., Michaelis, J., Becker, S., Dorff, H., Ebell, K., Gorodetskaya, I., Heinold, B., Kirbus, B., Lauer, M., Maherndl, N., Maturilli, M., Mayer, J., Müller, H., Neggers, R. A. J., Paulus, F. M., Röttenbacher, J., Rückert, J. E., Schirmacher, I., Slättberg, N., Ehrlich, A., Wendisch, M., and Crewell, S.: Contrasting extremely warm and long-lasting cold air anomalies in the North Atlantic sector of the Arctic during the HALO-(AC)[3] campaign, Atmos. Chem. Phys., 24, 8007–8029, https://doi.org/10.5194/acp-24-8007-2024, 2024.

**This paper describes the data collected from the three aircraft. Ground based observations from the AWIPEV base are mentioned in line 75 to 82, but no further reference to data sets is given, except for the radiosonde observations. It would be useful to add further references, at least for the mentioned tethered balloon data and the ground based remote sensing data sets.**

Indeed, our intention with this paper was to restrict to the aircraft data set, which includes a huge amount of data measured by a multitude of instruments. In the context of HALO-$(AC)^3$ additional observations are available such as ground-based and balloon measurements at AWIPEV. We decided not to include these data and their description in our paper, also because there are plans to publish and present the data sets separately. Combining the aircraft, balloon, and ground-based data would have clearly exceeded the scope of this paper.

However, as suggested by the reviewer, we have added relevant references providing information on these additional data sets. The corresponding text was changed to:

> *"The tethered balloon BELUGA (Balloon-born moduLar Utility for profilinG the lower Atmosphere, Egerer et al., 2019) collected vertical profile data in the atmospheric boundary layer and the lower free troposphere from the ground to about 1 km height using sophisticated turbulence, radiation, and aerosol instrumentation (e.g., Pilz et al., 2023). An overview of the balloon-born observations conducted during HALO–$(AC)^3$ is given by Lonardi et al. (2024). During March and April 2022, the frequency of the regular radiosonde launches at AWIPEV was increased to six-hourly intervals. These data are published by Maturilli (2022a, b). A long-term data set of ground-based cloud remote sensing observations at AWIPEV is published by Chellini et al. (2023)."*

Chellini, G., Gierens, R., Ebell, K., Kiszler, T., Krobot, P., Myagkov, A., Schemann, V., and Kneifel, S.: Low-level mixed-phase clouds at the high Arctic site of Ny-Ålesund: a comprehensive long-term dataset of remote sensing observations,

Earth System Science Data, 15, 5427–5448, https://doi.org/10.5194/essd-15-5427-2023, 2023.

Lonardi, M., Akansu, E. F., Ehrlich, A., Mazzola, M., Pilz, C., Shupe, M. D., Siebert, H., and Wendisch, M.: Tethered balloon-borne observations of thermal-infrared irradiance and cooling rate profiles in the Arctic atmospheric boundary layer, Atmos. Chem. Phys., 24, 1961–1978, https://doi.org/10.5194/acp-24-1961-2024, 2024.

Maturilli, M.: High resolution radiosonde measurements from station Ny-Ålesund (2022-03), PANGAEA, https://doi.org/10.1594/PANGAEA.944406, in: Maturilli, M (2020): High resolution radiosonde measurements from station Ny-Ålesund (2017-04 et seq). Alfred Wegener Institute - Research Unit Potsdam, PANGAEA, https://doi.org/10.1594/PANGAEA.914973, 2022a.

Maturilli, M.: High resolution radiosonde measurements from station Ny-Ålesund (2022-04), PANGAEA, https://doi.org/10.1594/PANGAEA.944409, in: Maturilli, M (2020): High resolution radiosonde measurements from station Ny- Ålesund (2017-04 et seq). Alfred Wegener Institute - Research Unit Potsdam, PANGAEA, https://doi.org/10.1594/PANGAEA.914973,1060, 2022b.

**An essential part of the project is the data set of lagrangian matches of air mass trajectories, which have been probed twice, thereby allowing to determine the temporal air mass development. To my understanding the data sets Kirbus et al. 2024a and Kirbus et al. 2024b provide matches of air masses probed by the HALO aircraft only. Wouldn't it by useful to have also matches between air masses probed by HALO and Polar 5, or 6, at least for the drop sonde data? If not, this should be more clearly mentioned in chapter 3.1.5 and especially in the conclusions (line 730f)**

This is correct, the two data sets only account for Lagrangian matches achieved with HALO. We tried to identify the matches including Polar 5 and Polar 6, but for several reasons this turned out to be not useful. (i) The flight patterns of Polar 5 and Polar 6 were designed to be collocated with HALO most of the time, e.g., along the standard leg, for having simultaneous in situ measurement. That is why, Lagrangian matches from this leg are already covered in the HALO-HALO matches. (ii) Matching observations from HALO and Polar 5 would have the disadvantage that Polar 5 only covered the lowest 3 km of the atmosphere. (iii) Matches between Polar 5 and Polar 6 (without HALO) were also not considered in the flight planning and would only cover small scales, e.g., just a few minutes covering a rather limited area, too limited to observe significant air mass transformation. When searching the quasi-Lagrangian matches, a constraint of a minimum threshold of 1 hour between

first and second sampling was applied. This limit was mostly not exceeded by both Polar aircraft.

In the revised manuscript we add this statement:

> *"For a reasonable analysis of air mass transformations, the data sets include only matches with a minimum threshold of one hour between the first and second sampling. This limits the analysis to HALO flight tracks. The flight tracks of Polar 5 and Polar 6 did not cover such long distances along the trajectories."*

**A minor remark on chapter 5.3, line 700:**

**"Can be applied to validate satellite products" is only a very general statement. Could you give an example, or a reference, of what could be learned from this combination of passive sensors data?**

That's true, we had not been very specific here. What we have in mind here is, that satellite products based on passive remote sensing observations typically rely on a combination of data from solar and thermal-infrared spectral channels (e.g., visible channels for cloud optical thickness, near-infrared channels for cloud effective radius and cloud phase, and thermal-infrared channels for cloud top altitude). With the combination of instruments operated on HALO and Polar 5, we can mimic such typical satellite observations in all relevant spectral bands. As an example, we currently apply the retrieval algorithm of the EarthCARE multispectral imager (MSI) to airborne observations from HALO-(AC)[3]. Using the high spatial resolution of the airborne observations, we can learn how the coarser spatial resolution of the satellite measurements affect the retrieved cloud properties. Similar studies using active radar remote sensing observations to evaluate CloudSat retrieval products have been published by Schirmacher et al. (2023). Passive microwave of sea ice emissivity from airborne and satellite observations are investigated by Risse et al. (2024)

In the revised manuscript we are more specific with the aspects of satellite remote sensing:

> *"By combining data from these different spectral ranges, the airborne observations can mimic the most common satellite imaging spectrometers with a superior spatial resolution. Satellite retrieval algorithms can then be applied to the airborne data to quantify the impact of spatial averaging on the satellite products of surface and cloud properties."*

Schirmacher, I., Kollias, P., Lamer, K., Mech, M., Pfitzenmaier, L., Wendisch, M., and Crewell, S.: Assessing Arctic low-level clouds and precipitation from above – a radar perspective, Atmos. Meas. Tech., 16, 4081–4100, https://doi.org/10.5194/amt-16-4081-2023, 2023.

Risse, N., Mech, M., Prigent, C., Spreen, G., and Crewell, S.: Assessing sea ice microwave emissivity up to submillimeter waves from airborne and satellite observations, The Cryosphere, 18, 4137–4163, https://doi.org/10.5194/tc-18-4137-2024, 2024.

**Line 109 / Figure 2: Figure 2 could be omitted as the main information is well described in the text. Relative flight duration given in % does not seem to be very meaningful to this reviewer.**

We decided to keep Figure 2 as it includes additional useful information than stated in the text. Also, visual comparison of the contribution by the individual surface types helps the reader to remember these important numbers without searching in the text. The relative flight duration in % was chosen to make the data of the three aircraft comparable. HALO has an almost three times longer endurance compared to Polar 5 and Polar 6. Thus, absolute hours would have mostly shown this mismatch. The total flight hours are presented in the figure legend. That's why we think, it is easy to convert into flight hours if needed.

**Some specific remarks and suggestions:**

**Author list: C. Lüpkes is of AWI not LIM ?**

Of course, he is! Sorry for this typo and thanks for your careful reading.

**Line 63: wording of "balloon-borne observations in area around Svalbard"**

Was changed by adding a "the".

**Line 74: wording of "... during one single and along successive flights"**

Was changed into:

*"HALO providing the large scale view following air masses with a quasi-Lagrangian flight strategy."*

**Line 83 and 84: any references available to descriptions of the ISLAS and ACAO campaigns?**

There are no general campaign overviews of ISLAS and ACAO available yet. However, for ISLAS we included a corresponding doi-citable webpage, where project reports and publications are available. For ACAO, a recent study also providing a brief overview of the measurements is now included in the revised text of our paper.

*Sodemann, H.: Isotopic links to atmospheric water's sources - ISLAS, European research council, Grant agreement ID: 773245, https://doi.org/10.3030/773245, 2018.*

*Raif, E. N., Barr, S. L., Tarn, M. D., McQuaid, J. B., Daily, M. I., Abel, S. J., Barrett, P. A., Bower, K. N., Field, P. R., Carslaw, K. S., and Murray, B. J.: High ice-nucleating particle concentrations associated with Arctic haze in springtime cold-air outbreaks, EGUsphere, 2024, 1–38, https://doi.org/10.5194/egusphere-2024-1502, 2024.*

**Line 94 should read "between two fixed waypoints"**

Thanks! We corrected this.

**Line 131,132: wording "were operated almost identical setup"**

Thanks! We corrected this.

**Line 133: should read "devices were extended"**

Thanks! We corrected this.

**Line 215: I think you want to say that in the stratosphere the accuracy of humidity measurements is low.**

Thanks! We corrected this.

**Line 269: "two versions of SMART": what kind of 2 versions are these? One for each plane? What is their difference? See also line 275 where a 1-2 sentence description of SMART would be helpful.**

Yes, on both planes one system was operated. Maybe "versions" was misleading, as the main components are almost identical. We omitted "versions" and added the following sentence:

*"The two systems, one installed on HALO and one on Polar 5, utilize identical types of grating spectrometers and optical inlets. They differ only in the implementation of the horizontal stabilization."*

**Line 275: "two types of grating spectrometers": which 2 types are these? see comment in line 269**

Thanks for pointing at this missing information. However, to avoid going into do many details of the system, we decided to remove "two types". The specifications of the final data set are described, what we think is sufficient here. More details are not necessary to understand the published data. Those who are interested in these details can use the provided references.

**Lines 278 and 279: should read "wavelengths"**

Thanks! We corrected this.

**Line 295: double word "depolarization"**

Thanks! We corrected this.

**Line 412ff: there are two Moser et al. 2023 references in the literature section. Please be specific, which reference is used where (should be 2023a and 2023b)**

Thanks! We solved this bibtex problem.

**Line 480: double word "to"**

Thanks! We corrected this.

**Line 535: "N. and Y., 2010" as well as in reference list: this should be the reference "Moteki and Kondo, 2010" You also need to correct it in the reference list.**

Thanks! We solved this bibtex problem.

**Line 547: wording "which not always sufficient,"**

Thanks! We corrected this.

**Line 560: wording "to investigating"**

Thanks! We corrected this.

**Line 603: wording: "With the exception of some instruments available in compressed ASCII format, "**

Thanks! We corrected this.

**Figure 6 (before Line 680): Please add in the figure caption something like: "Blue line in panel (b) gives flight altitude of Polar 6 for in situ sampling."**

Thanks! We added this.

**Line 731: please write "the meteorological data transferred to GTS"**

Thanks! We added this.

**Further changes:**

We checked and corrected grammar and wording in some instances.

The unified HAMP data set (Dorff et al., 2023) was updated for new calibration coefficients of the passive microwave radiometer. The new data (Dorff et al. 2024) are published as a revised data set on PANGAEA. The reference was exchanges in the manuscript to guide readers directly to the new revised data.

Dorff, H., Aubry, C., Ewald, F., Hirsch, L., Jansen, F., Konow, H., Mech, M., Ori, D., Ringel, M., Walbröl, A., Crewell, S., Ehrlich, A., Wendisch, M., and Ament, F.: Unified Airborne Active and Passive Microwave Measurements over Arctic Sea Ice and Ocean during the HALO-(AC)[3] Campaign in Spring 2022 (v2.7), https://doi.org/10.1594/PANGAEA.974108, 2024.

---

## Author Comment (AC2)

**Reply to RC2: 'Comment on essd-2024-281', Anonymous Referee #2, 27 Oct 2024**

Thanks for the detailed and constructive review. The comments of the reviewer have been helpful to improve the manuscript. We are especially thankful for the suggestion to emphasize the data sets of other HALO-(AC)³ activities, which significantly increased the value of the manuscript for potential readers.

Our detailed replies on the reviewers' comments are elaborated below. The reviewers' comments are given in bold, while our replies are in regular roman letters. Citations from the revised manuscript are given in italic fonts.

**General comments:**

**1. Sect. 2.1 of general setup: The authors mentioned the ground-based and balloon-borne observations, however, the associated instrumentation, data processing, archive, and publication are not provided. Could authors provide more information related to these observations?**

It is true, that we only mention these data. Indeed, our intention with this paper was to restrict to the aircraft data set, which includes a huge amount of data measured by a multitude of instruments. In the context of HALO-(AC)³ additional observations are available such as ground-based and balloon measurements at AWIPEV. We decided not to include these data and their description in our paper, also because there are plans to publish and present the data sets separately. Combining the aircraft, balloon, and ground-based data would have clearly exceeded the scope of this paper.

However, as suggested by the reviewer, we have added relevant references including those where details on the instrumentation are published and made clearer, which of these references provide the data sets. The corresponding text was changed to:

> *"The tethered balloon BELUGA (Balloon-born moduLar Utility for profilinG the lower Atmosphere, Egerer et al., 2019) collected vertical profile data in the atmospheric boundary layer and the lower free troposphere from the ground to about 1 km height using sophisticated turbulence, radiation, and aerosol instrumentation (e.g., Pilz et al., 2023). An overview of the balloon-born observations conducted during HALO–(AC)³ is given by Lonardi et al. (2024). During March and April 2022, the frequency of the regular radiosonde launches at AWIPEV was increased to six-hourly intervals. These data are published by Maturilli (2022a, b). A long-term data set of ground-based cloud remote sensing observations at AWIPEV is published by Chellini et al. (2023)."*

Chellini, G., Gierens, R., Ebell, K., Kiszler, T., Krobot, P., Myagkov, A., Schemann, V., and Kneifel, S.: Low-level mixed-phase clouds at the high Arctic site of Ny-Ålesund: a comprehensive long-term dataset of remote sensing observations, Earth System Science Data, 15, 5427–5448, https://doi.org/10.5194/essd-15-5427-2023, 2023.

Lonardi, M., Akansu, E. F., Ehrlich, A., Mazzola, M., Pilz, C., Shupe, M. D., Siebert, H., and Wendisch, M.: Tethered balloon-borne observations of thermal-infrared irradiance and cooling rate profiles in the Arctic atmospheric boundary layer, Atmos. Chem. Phys., 24, 1961–1978, https://doi.org/10.5194/acp-24-1961-2024, 2024.

Maturilli, M.: High resolution radiosonde measurements from station Ny-Ålesund (2022-03), PANGAEA, https://doi.org/10.1594/PANGAEA.944406, in: Maturilli, M (2020): High resolution radiosonde measurements from station Ny-Ålesund (2017-04 et seq). Alfred Wegener Institute - Research Unit Potsdam, PANGAEA, https://doi.org/10.1594/PANGAEA.914973, 2022a.

Maturilli, M.: High resolution radiosonde measurements from station Ny-Ålesund (2022-04), PANGAEA, https://doi.org/10.1594/PANGAEA.944409, in: Maturilli, M (2020): High resolution radiosonde measurements from station Ny- Ålesund (2017-04 et seq). Alfred Wegener Institute - Research Unit Potsdam, PANGAEA, https://doi.org/10.1594/PANGAEA.914973,1060, 2022b.

**2. Lines 42-43: The authors mentioned that "the HALO–(AC) 3 aircraft campaign was designed to combine different aircraft platforms and aiming for observations in a quasi-Lagrangian approach". Lines 50-52: The authors also mentioned that "CAOs were characterized in their initial stage close to the sea ice edge with the Polar 5 and Polar 6 research aircraft and in a quasi-Lagrangian perspective jointly with HALO".**

**However, in Sect. 3.1.5 and Table 4, the analysis approach and data publication of quasi-Lagrangian matches are described only for HALO aircraft flights. Are quasi-Lagrangian matches analyzed between HALO and Polar 5, or 6 flights? If yes, could authors provide more information related to such as its data publication?**

This is correct, the two data sets only account for Lagrangian matches achieved with HALO. We tried to identify the matches including Polar 5 and Polar 6, but for several reasons this turned out to be not useful. (i) The flight patterns of Polar 5 and Polar 6 were designed to be collocated with HALO most of the time, e.g., along the standard leg, for having simultaneous in situ

measurement. That is why, Lagrangian matches from this leg are already covered in the HALO-HALO matches. (ii) Matching observations from HALO and Polar 5 would have the disadvantage that Polar 5 only covered the lowest 3 km of the atmosphere. (iii) Matches between Polar 5 and Polar 6 (without HALO) were also not considered in the flight planning and would only cover small scales, e.g., just a few minutes covering a rather limited area, too limited to observe significant air mass transformation. When searching the quasi-Lagrangian matches, a constraint of a minimum threshold of 1 hour between first and second sampling was applied. This limit was mostly not exceeded by both Polar aircraft.

In the revised manuscript we add this statement:

> *"For a reasonable analysis of air mass transformations, the data sets include only matches with a minimum threshold of one hour between the first and second sampling. This limits the analysis to HALO flight tracks. The flight tracks of Polar 5 and Polar 6 did not cover such long distances along the trajectories."*

**If it is available, could you also provide an example figure of quasi-Lagrangian matches?**

Thanks for this suggestion. We agree, that an example figure helps to understand the nature of the quasi-Lagrangian matches. As a representative example, we chose RF03 where a warm air intrusion was captured by HALO (see new Figure 4 below). Here it is obvious, which part of the flight includes the matches and how wind shear affects the vertical distribution of the locations. In the manuscript we added:

> *"Figure 4 illustrates the horizontal and vertical location of quasi-Lagrangian matches that occurred within the research flight number 3 (RF03) performed on 13 March 2024. All locations where the air mass was sampled the first time (start points) are distributed on the zigzag leg directed northward. On the return flight leg these air masses were samples a second time (end point). In total, seven quasi-Lagrangian matches (indicated by numbers in Fig. 4) were identified."*

and

> *"Vertically, trajectories were initialized at all altitudes below HALO with a 5 hPa resolution to consider wind shear, ascend, and descend of air masses (see Wendisch et al., 2024). This is obvious by the distribution of the start/end points in Fig. 4b where the branches of quasi-Lagrangian matches are diffusing and not restricted to a fixed vertical column."*

[Figure]

Figure 4. Visualization of quasi-Lagrangian matches for the HALO RF03 on 13 March 2022. The map in a) shows the flight track of HALO overlaid on ERA5 data of sea ice concentration and integrated water vapor transport (gray arrows). In b), a vertical cross section of radar reflectivity is shown. The first sampling location (start points) of matches are indicated in both panels by orange dots, the second location (end points) by red dots. For the map in panel a) theses start/end points represent altitudes close to the surface and are connected by the trajectories. Number labels in both panels indicate the individual branches of start/end points.

**3. Section 3.5.3: The analysis method of INP filters is not provided in the manuscript, which is an important part of INP filter methodology. Additionally, the publication of the INP dataset is not provided. Could authors provide more information related to INP filters?**

> The analysis of the INP filter was not included by intention for several reasons. (i) the data prescription is focusing on the primarily measured quantities. In the case of the INP filters, these are physical samples of aerosol particles. These can not be published like all recorded data. (ii) the analysis of the filter is done offline and   different methods can be used, e.g., different freezing arrays for INP analysis (e.g., Chen et al. 2018, Hartmann et al. 2019) or scanning electron microscopy for particle morphology analysis (e.g., Seifried et al. 2021) or chemical composition analysis of the filtered particles (e.g., Kwiezinski et al. 2021). In our view, this analysis goes beyond the scope of the data description paper. Similar, other data sets, e.g. remote sensing data, can be postprocessed to derive additional data products afterwards.

To avoid the impression, that INP concentrations are published in the data set, we changed the title of the subsection to "*Polar 6 - Cloud condensation particles measurements and particle filter sampling*" and explained the potential applications of the filter samples:

> *"The High-volume flow aERosol particle filter sAmpler (HERA, Grawe et al., 2023) was deployed for collecting aerosol particles that can be used in offline laboratory analysis. For example, the samples can be used for deriving the ice nucleating particles (INP) concentrations (e.g., Chen et al., 2018; Hartmann et al., 2019), scanning electron microscopy for particle morphology analysis (e.g., Seifried et al., 2021) or chemical composition analysis of the filtered particles (e.g., Kwiezinski et al., 2021)."*

Chen, J., Wu, Z., Augustin-Bauditz, S., Grawe, S., Hartmann, M., Pei, X., Liu, Z., Ji, D., and Wex, H.: Ice-nucleating particle concentrations unaffected by urban air pollution in Beijing, China, Atmos. Chem. Phys., 18, 3523–3539, https://doi.org/10.5194/acp-18-3523-2018, 2018.

Hartmann, M., Blunier, T., Brügger, S., Schmale, J., Schwikowski, M., Vogel, A., Wex, H., and Stratmann, F.: Variation of Ice Nucleating Particles in the European Arctic Over the Last Centuries, Geophys. Res. Lett., 46, 4007–4016, https://doi.org/10.1029/2019GL082311,2019.

Seifried, T. M., Bieber, P., Kunert, A. T., Schmale, D. G., Whitmore, K., Fröhlich-Nowoisky, J., and Grothe, H.: Ice Nucleation Activity of Alpine Bioaerosol Emitted in Vicinity of a Birch Forest, Atmosphere, 12, 779, https://doi.org/10.3390/atmos12060779, 2021.

Kwiezinski, C., Weller, C., van Pinxteren, D., Brüggemann, M., Mertes, S., Stratmann, F., and Herrmann, H.: Determination of highly polar compounds in atmospheric aerosol particles at ultra-trace levels using ion chromatography Orbitrap mass spectrometry, J. Sep. Sci., 44, 2343–2357, https://doi.org/10.1002/jssc.202001048, 2021

**Specific comments:**

**Page 7, Line 133: "the cloud microphysical devices was extended…" should be "were extended".**

Thanks! We corrected this.

**Page 11, Line 180: It will be nice to provide which flight numbers are not available for the Polar 5 and Polar wind data.**

We added the research flight number for which wind data are missing:

*"Problems concerned especially data from Polar~5, where we cannot provide wind data from nine flights (RFs 08-16) while from Polar~6 wind data are missing from five flights (RFs 06-08, 12, 16)"*

**Page 12, Line 215: "where the accuracy of measurements is large" should be "where the uncertainty of measurements is large".**

Thanks! We corrected this.

**Page 14, Line 249: "Measurements during HALO–(AC)[3] are characterized…" should be "were characterized".**

Thanks! We corrected this.

**Page 14, Line 251: "the solar leakage effect of the pyrgeometer effecting the…" should be "affecting the…".**

Thanks! We corrected this.

**Page 15, Line 269: the authors mentioned that "Spectrally resolved downward (Polar 5, HALO) and upward (Polar 5 only) solar irradiance were measured using two versions of the Spectral Modular Airborne Radiation measurement sysTem (SMART)". What are the two versions of the SMART and the corresponding versions for HALO and Polar5? What are the differences between the two versions of the SMART?**

Yes, on both planes one system was operated. Maybe the term "versions" was misleading, as the main components are almost identical. We omitted "versions" and added the following sentence:

*"The two systems, one installed on HALO and one on Polar 5, utilize identical types of grating spectrometers and optical inlets. They differ only in the implementation of the horizontal stabilization."*

Differences in terms of measured radiative quantities (irradiance/radiance and upward/downward) are listed in Tables 2 and 3.

**Page 15, Line 284: The authors mentioned that "the Polar 5 data set was filtered for large SZA and aircraft pitch and roll angle". Why this filter criteria was not applied to the HALO dataset?**

The two systems have different mechanics to actively level the optical irradiance inlets into a horizontal position. On HALO, the performance of this stabilization is easier to obtain because the roll/pitch changes of HALO are weaker compared to Polar 5. HALO flies faster, higher and, therefore, more smoothly. This allows to meaningfully analyze measurements conducted under low Sun conditions. Still, the uncertainty is increased for irradiance measurements with HALO radiometers when flying close to the North Pole with extremely large solar zenith angles. As HALO flew frequently in these conditions, removing the data would reduce the amount of measurements significantly, which is not the case for Polar 5. Thus, we decided to not filter the HALO data beforehand.

In the revised manuscript we clarified this issue:

> *"HALO data were not filtered because the horizontal stabilization is more accurate and the amount of flight time in the highest latitudes with low Sun was higher."*

**Page 15, Line 295: "particle linear depolarization depolarization", duplicate "depolarization".**

Thanks! We corrected this.

**Page 16, Lines 297-298: "The backscatter profiles are extinction-corrected..." should be "were...".**

Thanks! We corrected this.

**Page 16, Lines 304-305: The authors mentioned "The data are considered of good quality if no flag is set, i.e., if the flag is zero. Note that for a non-zero flag, the data value itself is not replaced by a fill-value." What is the recommended treatment for non-zero-flag value? Nor reliable to use or?**

There is no general recommendation for the treatment of non-zero quality values. It depends on the type of analysis and the objectives of the data user. To keep as many options for analysis as possible, we used the flags and did not remove the data value. Here is one example: E.g. a cloud mask can be produced from less quality WALES data while retrieval of aerosol properties using data of the same quality would be significantly wrong. Similarly, data averaging might overcome these issues. This is only one example. Explaining all possibilities is not possible in the data paper. Instead, we added:

> *"Note that for a non-zero flag, the data value itself is not replaced by a fill-value because it still can be used for some analysis. Thus, potential users have to make sure to filter the data by help of the flags according to their objectives of the data analysis or retrieval."*

**Page 16, Line 308: "at 355 nm and 532 nm wavelength" ---> "wavelengths".**

Thanks! We corrected this.

**Page 20, Lines 451-452: The authors mentioned that "Due to the complexity of deriving microphysical properties from OAPs different solutions are included in the data sets, which can be selected depending on the focus of analysis". What do "different solutions" refer to?**

Sorry, we had not been very specific and precise on this important detail! The particle size distributions are provided for different definitions of the particle diameter of ice crystals, which have established in the literature. Due to the irregular shape of ice crystals, different solutions for quantifying the particle size are possible. In the dataset, the diameter for surface equivalent (Deq) and the diameter of a circumscribed circle (Dcc) are provided. As ice crystals may only be partly imaged by the limited sensor width, two solutions, one considering all images (index 0) and one considering only complete ice crystals (index 1) are analyzed. For the 2D-S imaging probe separate analysis of the horizontal and vertical camera are included in the data set.

Similarly, the computation of the effective diameter using the ratio of the moment of order 3 to the moment of order 2 is only correct for spherical particles, thus this formulation cannot be applied to ice crystals. For ice crystals, the ratio of the ice water content to the ice extinction coefficient (related to the total ice crystals cross section surface) needs to be applied but cannot directly measured by 2D imaging probes. Therefore, a mass diameter relationships need to be applied to compute the ice water content and to derive the effective diameter. The published data set provides effective diameter and IWC based on the two mass diameter relationship by Field et al. (2006) and Brown and Francis (1995).

These multitude of data versions, which all are reasonable, allow user to choose the most suitable data processing for their specific objective in the data analysis.

In the revised manuscript, we added this information:

*"The 2D-S data are published by Dupuy et al. (2024) and are provided separately for the horizontal and vertical viewing direction. Due to the complexity of deriving microphysical properties from OAPs, different solutions are included in the data sets, which can be selected depending on the focus of analysis. For ice crystal size, the diameter for surface equivalent (Deq) and the diameter of a circumscribed circle (Dcc) are provided. As ice crystals may only be partly imaged by the limited sensor width, two solutions, one considering all images (index 0) and one considering only complete ice crystals (index 1) are analyzed. For the IWC two mass diameter relationship by Field et al. (2006) and Brown and Francis (1995) were applied and both results are included in the data set. The effective diameter was computed based on the ratio of the IWC by the ice extinction which is related to the total ice crystal cross section as in Gayet et al. (2002)."*

**Page 21, Lines 469-470: "The respective enrichment factor (EF) needs to be applied to every instrument sampling CPRs". Could authors provide a reference or brief introduction about the calculation method of EF?**

We agree that more information is needed for reader to apply the enrichment factor to the data. The revised manuscript now includes the following explanation:

*"EF is defined by the ratio of the air volume flow in front of and inside the CVI inlet. The inlet sample flow inside the CVI was measured continuously and depends on the number of instruments connected to the CVI. The in front flow was determined from the aircraft speed relative to the air and the inlet diameter. Since the Polar 6 aircraft speed is rather low, EF values were rather low, ranging between 3 and 6, depending on the sample flow."*

**Page 21, Line 490: "the number concentration of particles larger than 10 nm wass" ---> "was".**

Thanks! We corrected this.

**Page 24, Line 560: " to investigating potential" ---> "to investigate"**

Thanks! We corrected this.

**Page 33, Line 726: "cold air outbreaks (COAs)" should be rephrased to "cold air outbreaks (CAOs)".**

Thanks! We corrected this.

**Further changes:**

We checked and corrected grammar and wording in some instances.

The unified HAMP data set (Dorff et al., 2023) was updated for new calibration coefficients of the passive microwave radiometer. The new data (Dorff et al. 2024) are published as a revised data set on PANGAEA. The reference was exchanges in the manuscript to guide readers directly to the new revised data.

Dorff, H., Aubry, C., Ewald, F., Hirsch, L., Jansen, F., Konow, H., Mech, M., Ori, D., Ringel, M., Walbröl, A., Crewell, S., Ehrlich, A., Wendisch, M., and Ament, F.: Unified Airborne Active and Passive Microwave Measurements over Arctic Sea Ice and Ocean during the HALO-(AC)[3] Campaign in Spring 2022 (v2.7), https://doi.org/10.1594/PANGAEA.974108, 2024.